# Data Integration of Critical Elements from Mine Waste in Mexico, Chile and Australia

Denys Villa Gomez [1,*] , Enrique Sáez Salgado [2] , Olivia Mejías [2], Aurora Margarita Pat-Espadas [3], Laura Alejandra Pinedo Torres [4], Laura Jackson [2] and Anita Parbhakar-Fox [2]

1   School of Civil Engineering, The University of Queensland, Brisbane 4072, Australia
2   W.H. Bryan Mining & Geology Research Centre, Sustainable Minerals Institute, University of Queensland, Brisbane 4068, Australia; e.saezsalgado@uq.edu.au (E.S.S.); o.mejiasgonzalez@uq.edu.au (O.M.); laura.jackson@uq.edu.au (L.J.); a.parbhakarfox@uq.edu.au (A.P.-F.)
3   CONACYT-UNAM Instituto de Geología, Estación Regional del Noroeste, Avenida Luis D. Colosio Esquina Madrid, Hermosillo 83000, Mexico; apespadas@geologia.unam.mx
4   Unidad Profesional Interdisciplinaria de Ingeniería Campus Zacatecas, Instituto Politécnico Nacional Zacatecas, Zacatecas 98160, Mexico; lauraalejandrapinedotorres@yahoo.com.mx
*   Correspondence: d.villagomez@uq.edu.au

**Abstract:** Due to an extensive history of mining activities common to Mexico, Chile and Australia there is a vast mine waste legacy. Whilst these wastes present ongoing challenges regarding their management, they may represent a source of elements supporting the transition towards a low carbon future. Hence, our study aims to demonstrate the value of establishing a chemical database from publicly available tailings data collated from the three countries to assess their potential as a secondary resource of elements classified as critical or with high economic relevance. Overall, 2976 data samples were identified, analysed and georeferenced from 159, 642 and 7 Mexican, Chilean and Australian deposits, respectively. Data analysis shows that Mexico has significant potential for Bi, Sb, W, In, Zn and Mo with outstanding values in Sonora State, while Chile has significant potential for Bi, Sb, W and Mo, mostly from northern to central regions and Zn to the south. Whilst data from Australia are still being compiled, the potential for Co was recognised. The research exposes that available information is insufficient and highlights the need for an international report or assessment code for mine waste that encourages resource recovery from these resources and circular economy practices.

**Keywords:** data analysis; mine waste; tailings; critical elements; circular economy

## 1. Introduction

In recent years, the demand and price of elements with low natural abundance and unequal terrestrial distribution (classified as critical) have spiked due to new emerging technologies, such as smartphones, electric vehicles and green energy generation combating climate change. These elements include, for example, cobalt, platinum group metals (PGMs) and rare earth elements (REEs) [1]. The demand for these elements has resulted in significant economic, geopolitical and environmental challenges. The COVID-19 pandemic has exacerbated all of these; therefore, more than ever, the alternatives for recovering these metals from secondary resources are subject to increased interest [2].

In order to keep up with the increasing demand for metals, the mining industry has shifted to the exploitation of lower grade ores as the availability of higher-grade ores becomes scarcer. Nonetheless, the latter not only reduces economic viability but also comes with increased environmental costs [3,4]. As a result, tailings and waste rock are produced at a rate of over 100 billion tons per year worldwide [5]. This is the case of Mexico, Chile and Australia, for which their mining and metallurgical activities have resulted in a vast waste legacy and, therefore, common challenges and opportunities for evaluating

mine tailings for its potential as a secondary resource of critical elements. Due to the geological heterogeneity of the rocks mined and the continuous flow processes used in mineral processing, tailings can contain large quantities of valuable elements, including those classified as critical. Critical metals are often produced as a byproduct of mining activities from extracting base metals such as copper, lead, nickel and zinc [6]. For example, indium, germanium and gallium are typical byproducts of zinc refinement [7–9].

The recovery of critical elements from tailings could enhance circular economy practices in the mining industry as one of the circular economy's core principles, including reducing and minimising resource use, and methods to achieve that goal include recycling and reuse of wastes [10]. However, challenges for the valorisation of tailings include the lack of data integration from different sources, with information regarding tailings characteristics (e.g., mineral and chemical composition) and quantities [11]. If documented, this information could drive technological development enabling targeted processes for critical element extraction from tailings [10].

The objective of this study was to integrate and analyse current public information available for mine tailing sites in Mexico, Chile and Australia in order to assess their potential as a secondary resource of critical elements. These countries were selected based on the terms of the project awarded by the Council on Australia Latin America Relations (COALAR), which aimed to create a common front between Mexico, Chile and Australia to progress in the development of sustainable solutions for the recovery of critical elements from mine tailing. For Mexico and Chile, the study compiles available data on the concentration of a wide variety of elements hosted in tailings from numerous previous publications and available information from national geological surveys. For Australia, the study compiles data from samples collected and analysed by members of this research group. The statistical analysis of the data gathered allows identifying potential recovery critical elements by country This study is part of an international collaboration between Mexico, Chile and Australia that has built an online platform allowing users to analyse tailings as a secondary source of critical elements. Through continual input and growth of this database, the intention is that it will enable governments, industry, and academia to contribute towards research and development efforts into secondary extraction of critical elements from tailings sites.

## 2. Materials and Methods

The study focusses on three main aspects for each country: (1) a review of mining history and information available on the characterisation and identification of tailings deposits, (2) data collection of the identified tailings deposits containing geochemical information and (3) a database analysis comparing the concentrations of critical elements against reported crustal abundance and evaluating trends.

### 2.1. Data Collection

The information on tailings in Mexico was obtained from an extensive literature review of dissertations, indexed publications in international and national journals, reports and government websites (supplementary material Table S1). It is worth mentioning that in 2016, sectorial funding was provided to develop a nationwide inventory of abandoned mine liabilities, but the results are still in progress [12]. However, the academic sector has made efforts to study sites in different locations by providing information on tailing composition collected through the abovementioned sources. The data collected from journal articles about tailings in Mexico were obtained by searching three servers: Web of Science, ScienceDirect and Scopus. The keywords used in the searching were combinations of the following words: abandoned mine tailings (i.e., 51 results in Web of Science) and abandoned mine Mexico (i.e., 146 results in Web of Science). Only the papers reporting information about the mineral composition and elemental analysis were considered from these results. The analytical methods used in the literature from which the data were obtained included Inductively Coupled Plasma Mass Spectrometry (ICP-MS) and Atomic

Absorption Spectrometry (AAS) for the quantification of metals and metalloids after total acid digestion or sequential extraction of solid samples. The main analytical techniques for mineralogy, and major elements determination was conducted by using X-ray Diffraction (XRD) and X-ray Fluorescence (XRF). The data extracted were compiled and analysed, and when necessary, the appropriate factor conversion was used to preset the data in the same units (i.e., ppm).

The information on tailings in Chile was obtained from the considerable published information about tailings deposits and characteristics (including mineral and elemental composition) at the SERNAGEOMIN website (supplementary material-Table S2). This information is summarised in the report titled "Datos de Geoquímica de Depósitos de Relaves de Chile" (Geochemical Data from Tailings Deposits of Chile) [13]. The website also has a geochemical database from the program "Geochemical Characterization of Tailings Deposits in Chile" originated in 2017 by the Department of Tailings Deposits. The report and database, updated until January 2020, contains more than 2000 geochemical results from samples taken by the department since 2015. In addition, the analytical methods are described on the SERNAGEOMIN laboratory website [14]. They include X-ray Fluorescence (XRF) to determine major oxides and trace elements and Inductively Coupled Plasma Mass Spectrometry (ICP-MS) for REE and other trace elements quantification. The specimens were sampled from three primary locations of the tailings, including buckets' surface, dam walls and downstream soil sediments [13].

Whilst attempts have been made to record all mine waste features across Australia [15], a similar compendium of tailings chemical data (as presented for Chile) does not yet exist. Instead, these data are mostly held in restricted company databases (i.e., not for public dissemination) and research theses. Invariably, geochemical data are inconsistent, e.g., different analytical labs and methods used and portable XRF results reported in place of assay data. In recognition of this, Geoscience Australia highlighted the need to undertake a dedicated sampling and analysis program to assess critical mineral contents within Australia's mineral deposits and from mine products (including tailings) [11]. This approach has been adopted by the Queensland State Government (through the Geological Survey of Queensland), and it is anticipated that other Australian State Governments will also adopt this practice, as critical metal exploration across the country continues into the next decade. In this study, we have solely focussed on using a robust dataset collected through the Queensland Government's New Economy Minerals Initiative (NEMI).

This database, established in 2020, contains information from seven mine sites in Australia, specifically the state of Queensland. Tailing samples were collected at sites including Capricorn Copper, Wolfram Camp, Mount Garnet, Rocklands, Horn Island and Mary Kathleen. These were sampled by using hand auger methods with materials collected to a maximum depth of 10 m. The characteristics and sample numbers of the sites studied are presented in Supplementary Materials (Table S3). A soil auger drill rig was used to collect samples at the Herberton tailings to a maximum depth of 12 m. All samples were dried and pulverised before geochemical analysis. Tailing samples were subjected to multi-element geochemical analyses at ALS Global™ in Brisbane, Australia. The elements and detection limits for each of the used analytical suites are presented in Supplementary Materials (Table S4).

### 2.2. Database Analysis

The collected data for the three countries was organised by deposit, with location information including site name, state and coordinates, general and mineralogical characteristics of the site or region, as well as the source of information. In the case of Mexico, the data comprised 60 elements, conformed by 8 rock-forming minerals major elements stated in percent; 38 trace elements in ppm; and 14 REEs, also expressed in ppm. For Chile, the geochemical data included 55 elements and compounds for each sample, including 12 major elements from rock-forming minerals, expressed as oxides percent; content of sulphur (S) in percent; 29 common trace elements (below 1%) in ppm; 14 rare earth elements (REEs);

and a simple sum also for the total of the major elements. For Australia the geochemical data included 62 elements covering all the major elements from rock-forming minerals and 13 oxide compounds expressed in ppm.

The identified sites from the collected data were also georeferenced. In the case of the Mexican and Chilean sites, this was carried out through Universal Transverse Mercator projection, using DATUM reference systems WGS84 zones 11N, 12N, 13N, 14N, 15N, 18S and 19S. Software Arcmap (version 10.4, ESRI Inc., Redlands, CA, USA) was used to transpose the layers of Mexican metallogenetic provinces and Chilean regions.

Database analysis was carried out comparing the concentrations of elements classified as critical against reported crustal abundance (Section 2.3) and evaluating trends using the statistics and data science software Stata® (version 17.0, StataCorp, College Station, TX, USA) and geoscience software ioGAS™ (version 7.4, REFLEX, Balcatta, Australia) (Reflex ioGAS: Data Analysis Software).

Prior to data analysis, the data containing elements under the detection limit was replaced by null values in order to not overestimate concentrations in tailings deposits, while the values of elements above the detection limit were kept at the maximum detected value reported. Considering the amount of information for each country, analyses were grouped by political-administrative regions for Chile, metallogenic provinces for Mexico [16] and by deposit tailings in the case of Australia (Supplementary Materials Table S1). Statistical analysis was performed for each of these groups per country, including mean, minimum and maximum values, as well as standard deviation. Finally, the results were plotted using Tukey log-plot graphs to include quartiles analysis and outlier values.

### 2.3. Classification of Critical Elements and Crustal Abundance

The critical elements considered in this study were the ones ranked as most critical by the European Union (EU) on the list compiled and updated in 2020 by the European Commission (EC), which lists raw materials considered critical and vulnerable to supply disruptions [17] (Table 1). Additionally, raw materials with high relative economic importance have also been included in the analysis, as well as chromium, considered as a critical element by the United States and Japan [18]. These elements are also listed as critical by Australia [19], while Mexico and Chile have not created their list yet.

**Table 1.** Table of critical elements and raw materials with economic importance considering the EU list, US List and Japan List and their crustal abundance.

| Critical Elements | | EU List [a] | USA List [b] | Japan List [c] | Crustal Abundance [d] (ppm) |
|---|---|:---:|:---:|:---:|:---:|
| Platinum-group metals | Iridium (Ir) | ✓ | ✓ | ✓ | 0.001 |
| | Palladium (Pd) | ✓ | ✓ | ✓ | 0.015 |
| | Platinum (Pt) | ✓ | ✓ | ✓ | 0.005 |
| | Rhodium (Rh) | ✓ | ✓ | ✓ | 0.001 |
| | Ruthenium (Ru) | ✓ | ✓ | ✓ | 0.001 |
| Rare earth elements | Praseodymium (Pr) | ✓ | ✓ | ✓ | 9.2 |
| | Neodymium (Nd) | ✓ | ✓ | ✓ | 41.5 |
| | Cerium (Ce) | ✓ | ✓ | ✓ | 66.5 |
| | Terbium (Tb) | ✓ | ✓ | ✓ | 1.2 |
| | Dysprosium (Dy) | ✓ | ✓ | ✓ | 5.2 |
| | Yttrium [e] (Y) | ✓ | ✓ | ✓ | 33 |
| Other critical elements | Antimony (Sb) | ✓ | ✓ | ✓ | 0.2 |
| | Beryllium (Be) | ✓ | ✓ | ✓ | 2.8 |
| | Bismuth (Bi) | ✓ | ✓ | | 0.0085 |
| | Cobalt (Co) | ✓ | ✓ | ✓ | 25 |
| | Chromium (Cr) | | ✓ | ✓ | 102 |
| | Gallium (Ga) | ✓ | ✓ | ✓ | 19 |
| | Germanium (Ge) | ✓ | ✓ | ✓ | 1.5 |

**Table 1.** *Cont.*

| Critical Elements | | EU List [a] | USA List [b] | Japan List [c] | Crustal Abundance [d] (ppm) |
|---|---|:---:|:---:|:---:|:---:|
| | Hafnium (Hf) | ✓ | ✓ | | 3 |
| | Indium (In) | ✓ | ✓ | ✓ | 0.25 |
| | Lithium (Li) | ✓ | ✓ | ✓ | 20 |
| | Magnesium (Mg) | ✓ | ✓ | ✓ | 23,300 |
| | Niobium (Nb) | ✓ | ✓ | ✓ | 20 |
| | Phosphorus (P) | ✓ | | ✓ | 1050 |
| | Scandium (Sc) | ✓ | ✓ | | 22 |
| | Silicon [f] (Si) | ✓ | | | 282,000 |
| | Strontium (Sr) | ✓ | | ✓ | 370 |
| | Tantalum (Ta) | ✓ | ✓ | ✓ | 2 |
| | Titanium (Ti) | ✓ | ✓ | ✓ | 5600 |
| | Tungsten (W) | ✓ | ✓ | ✓ | 1.25 |
| | Vanadium (V) | ✓ | ✓ | ✓ | 120 |
| Raw materials with high relative economic importance | Iron (Fe) ore | ✓ | | | 56,300 |
| | Aluminium (Al) | ✓ | ✓ | | 82,300 |
| | Nickel (Ni) | ✓ | ✓ | ✓ | 84 |
| | Zinc (Zn) | ✓ | ✓ | ✓ | 70 |
| | Molybdenum (Mo) | ✓ | | ✓ | 1.2 |
| | Lanthanum (La) | ✓ | ✓ | ✓ | 39 |

[a] https://rmis.jrc.ec.europa.eu/uploads/CRM_2020_Report_Final.pdf (accessed date: 2 November 2021), [b] https://www.federalregister.gov/documents/2021/11/09/2021-24488/2021-draft-list-of-critical-minerals (accessed date: 10 January 2021), [c] https://www.industry.gov.au/sites/default/files/2019-03/australias-critical-minerals-strategy-2019.pdf (accessed date: 10 January 2021), [d] https://courses.lumenlearning.com/geology/chapter/reading-abundance-of-elements-in-earths-crust/#footnote-324-1 (accessed date: 2 January 2021), [e] as oxide, [f] metal.

In order to determine which critical elements could be considered for recovery due to their concentration in the tailing's deposits, the estimated abundance of these elements in the earth's crust in ppm was used as the threshold baseline (Table 1).

## 3. Results and Discussion

### 3.1. Mining, Tailings and Potential Recovery of Critical Elements in Mexico

Mining in Mexico dates back to the pre-Hispanic period when mines were established for Au and Ag extraction during the colonial era [20]. Currently, the mining-metallurgical sector contributes 2.3% of Mexico's gross domestic product, positioning it amongst the 14 top producers of 22 minerals worldwide, e.g., silver (first); fluorite (second); celestite, sodium sulphate and wollastonite (third); molybdenum, bismuth and lead (fifth); cadmium, magnesium sulphate and zinc (sixth); gold, gypsum and barite (eighth) [21]. At abandoned mine sites in the country, there are ongoing environmental management issues. However, it is unknown how many sites contain mine waste and tailings deposits (known as "*jales*" derived from the Náhuatl "*xalli*", which means fine sands [22]). It was not until 1992 that formal environmental legislation was included in the Mexican mining law [23]. Nowadays, some regulations govern exploration, extraction and closure in the mines [24].

Although efforts have been made to develop an inventory of tailings in the country [12,25], there is still a lack of integrated chemical and mineralogical information on these sites, thus highlighting the relevance of this study. The Mexican government has initiated the development of a preliminary inventory of tailings dams, which aims to facilitate access to information and the construction of public policies to manage these mining waste. To date, the inventory has identified 585 tailings dams, with some mineral information entered but limited geochemical information available [25]. In terms of current tailings storage impoundment, the Global Tailings Portal launched in 2020 reports 38 existing tailings dams in Mexico declared by mining companies with a total volume of over 400 million m$^3$.

The information about these tailing storage facilities includes location, company, dam type, height, volume and risk, among other factors [26].

In addition, the Mexican government has also released a national inventory of contaminated sites, which aims to promote remediation actions in contaminated sites to contribute to the population's wellbeing and strengthen the regulatory framework of contaminated sites. Up to 2016, 623 contaminated sites were registered, but no additional information regarding metals concentrations and characteristics is reported [27].

Regarding studies reporting the potential recovery of critical elements from mine waste, the information is very scarce, since most of the papers are focused on metals of environmental concern. However, from this review, only two cases are documented on that possibility and are in the early stages of feasibility assessment [28,29], while a recent study explored the concentrations of rare earth elements to determine their variations in samples, such as sediments, tailings and ash, from a mining district [30].

### 3.2. Mining, Tailings and Potential Recovery of Critical Elements in Chile

Chile has a long history of base-metal and world-class mining that started in the first decade of the twentieth century. The mining industry in Chile makes up 12.5% of the gross domestic product. It is the fifth producer of silver with 6.4% and the world's second-largest lithium producer with 26.5% [31]. In 2020, copper and molybdenum productions reached 5.77 million and 59,319 tonnes, respectively, positioning Chile as the world's leading copper producer with 28.5% of the total world production and the second producer of molybdenum with 20.2%. As such, over 530 million tonnes of tailings are produced per year from these mining industries, figures rising as a consequence of the decrease in the grade of copper, which increases the number of processed ores and, consequently, results in greater tailing deposits production.

According to the latest information in 2020 by the department of tailings deposits within the national geology and mining service (SERNAGEOMIN), the country has 757 tailings deposits [13] that are distributed from the region of Tarapacá in the north to Aysén in the south. Northern and central Chile regions have the most significant number of tailings deposits. The Coquimbo Region hosts 51.4% of them related to several small mineral processing plants, generating small tailing deposits [32]. The south of Chile hosts only 1.2% of the tailings in the Aysén Region; however, a study development in 2018 by Babel et al. [33] showed a high amount of coarse and liberated sphalerite particles in a southern zinc tailing. According to their operational status, 61.7% of the tailing deposits are inactive, 22.9% are classified as abandoned, 14.8% are currently active and 0.7% corresponds to tailings storage facilities under construction. The most significant numbers of tailings facilities are distributed in the north and central regions of the country. The Coquimbo Region hosts 51.4%, followed by the areas of Atacama (22.2%), Valparaiso (10.6%) and Antofagasta (6.9%), whilst the rest of the national locations have below 5% of the count of these facilities. In terms of current tailings storage impoundment, the total is 11.2 billion tonnes, representing 43% of the authorised storage. Antofagasta Region comprises the highest tailing storage tonnage (4.3 billion tonnes), followed by regions Metropolitana (2.6 billion tonnes), Coquimbo (1.3 billion tonnes) and Atacama (1.2 billion tonnes). Considering only active deposits, current mining tailings production is mainly concentrated in the north with 75% of the total, while 25% is produced in the central zone. Existing active impoundments represent a total of 10.6 billion tonnes, which corresponds to 94% of the current total tailing storage.

Although studies report the potential recovery of critical elements from mine waste [34–36], such initiatives are all in the early stages of feasibility assessment. It is worth mentioning that Minera Valle Central (MVC), located in the O'Higgins Region in central Chile produces Mo from old and new tailings, and in the words from its own website company, "It is the only company from all over the world which produces copper and molybdenum through tailings treatment" [37]. MVC daily reprocess approximately 200 kt of tailings mainly coming from El Teniente mine, a Cu-Mo porphyry deposit. In addition to the

aforementioned example, no other companies are reprocessing tailings for critical elements. Nevertheless, the economic potential of recovering vanadium, cobalt, rare earth elements and antimony from tailings from copper extraction has been reported [34].

### 3.3. Mining, Tailings and Potential Recovery of Critical Elements in Australia

The mining industry in Australia makes up 10.4% of the gross domestic product and is recognised as one of the world's largest exporters of coal, iron ore, lead, zinc and alumina [38]. Australia is also one of the world's top five producers of rutile, zircon, lithium, gold, manganese, nickel, diamonds, silver and copper [38]. Consequently, the mining industry in Australia produces large volumes of mine waste across a range of commodities, as tailings, waste rock and pyrometallurgical wastes.

Due to recent changes in ore price driven by the international push towards a low carbon future, a change in demand for metals required supporting green technologies. Consequently, there is an increased interest in critical metals for batteries with attitudes towards secondary sources of the metals, such as existing wastes including those from mining, recognised. Slowly, attitudes towards mine wastes are changing with some companies now recognising the potential metal resource they may hold, and the opportunity to unlock this will help the country meet targets set by the Federal Government for the Circular Economy to grow to a AUD 26 billion industry by 2025. Considering that the amount of inactive and unused mine sites across Australia is estimated at 80,000, appreciably, there is a significant number and range of sites where mine waste could host economic concentrations of critical minerals (Figure 1).

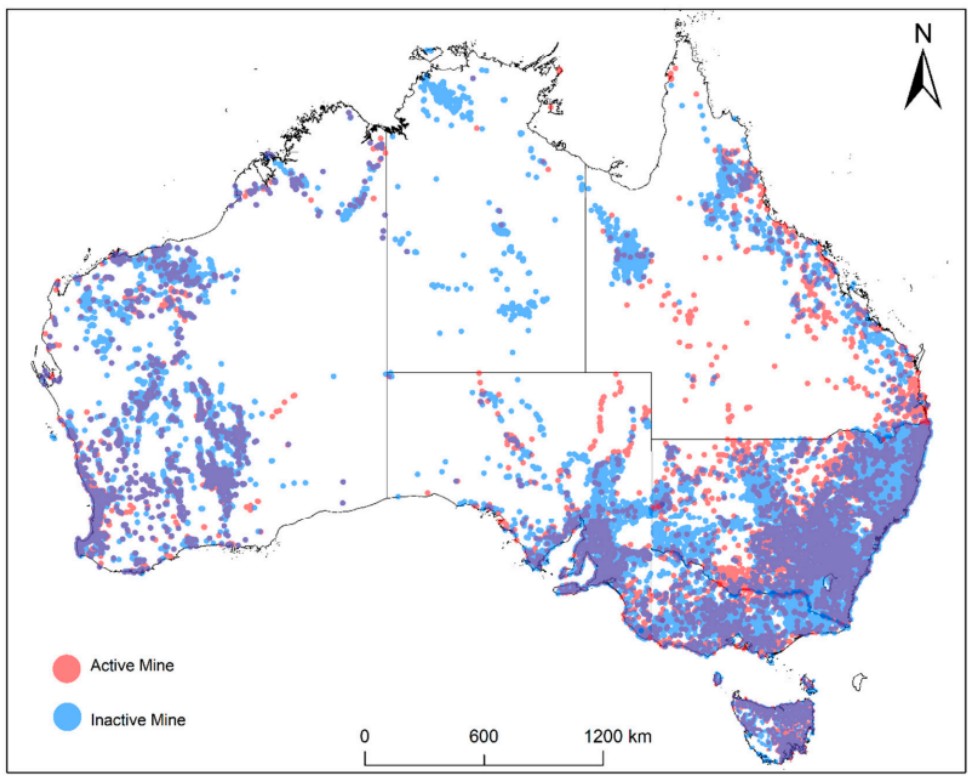

**Figure 1.** National overview of active and inactive mines across Australia [15].

Unlike Mexico and Chile, reprocessing of mine waste, including tailings, is already a business proposition that is increasingly being adopted in Australia [39]. For example, the Hellyer mine in Tasmania is actively recovering Pb-Zn (Au-Ag) from mine tailings. In Queensland, at the Century Mine, Zn is being recovered, after the mine closed in late 2015, and was then sold to a junior company named New Century. The re-treatment of tailings at Century commenced in August 2018 in order to extract residual zinc. It has been

estimated that there are 77 million tonnes of pre-crushed 3% zinc ore in the tailings dam, which allows for six and a half years of operation [40]. Other new projects include the Mt Carbine mine (W recovery; 2 Mt at 0.10% $WO_3$), Mt Morgan (10 Mt at 1.1 g/Au) and Tick Hill (630,000 t at 1.08g/t Au).

Motivated by these successes, increasing focus on critical metal recovery in Australia has been gaining traction with mining companies and state governments in particular. To provide context, in Queensland, there are at least 40 significant metalliferous mining operations producing mine waste streams containing unknown quantities of new economy metals. Additionally, there are 120 state-managed abandoned mines [41]. These sites can contain reactive sulfide-rich mine waste with associated acid and metalliferous drainage risks. The ongoing management of these sites is costly; therefore, the potential content of critical elements opens the opportunity to economically rehabilitate these sites through reprocessing waste [41].

*3.4. General Analysis of the Collected Data*

Overall, 2976 data samples were identified, analysed and georeferenced (Figure 2), with full datasets available in Supplementary Materials (Tables S1 and S2). Chilean samples comprised the majority of the samples obtained (72.6%) followed by Australia (17.5%) and Mexico (9.8%), with information sourced from a public database, from self-collected samples and from literature, respectively. As observed, this was a major study from collected data about tailing sites that provides a quantitative assessment of critical elements for potential recovery.

A summary of the total number of identified deposits and number of samples, average concentrations of critical elements and potential ratio (i.e., the average of an element divided by its crustal abundance) is shown in Table 2. For Mexico, a total of 274 data samples were obtained from Mexican tailing deposits corresponding to 159 tailings deposits. From that total, over 95% of the information collected only reported elements classified as toxic (i.e., As, Ba, Be, Cd, $Cr^{+6}$, Hg, Ni, Ag, Pb, Se, Tl and V) by the Mexican guidelines for maximum permissible limits of these elements in soil (NOM-147–SEMARNAT/SSA1-2004). Despite this and the fact that the database was the smallest among the three countries, 27 critical elements and raw materials with high relative economic importance were identified, which highlights the need to enrich this database to state the real potential for recovery of critical elements. For Chile, a total of 2032 data samples were obtained from Chilean tailing deposits corresponding to 642 out of a total of 757 tailings deposits registered by SERNAGEOMIN. From that total, only 21 critical elements or raw materials with high relative economic importance were identified. In the case of Australia, from the total of 490 samples taken from seven sites located in the north of Queensland (Figure 2), 16 elements critical elements or raw materials with high relative economic importance were identified.

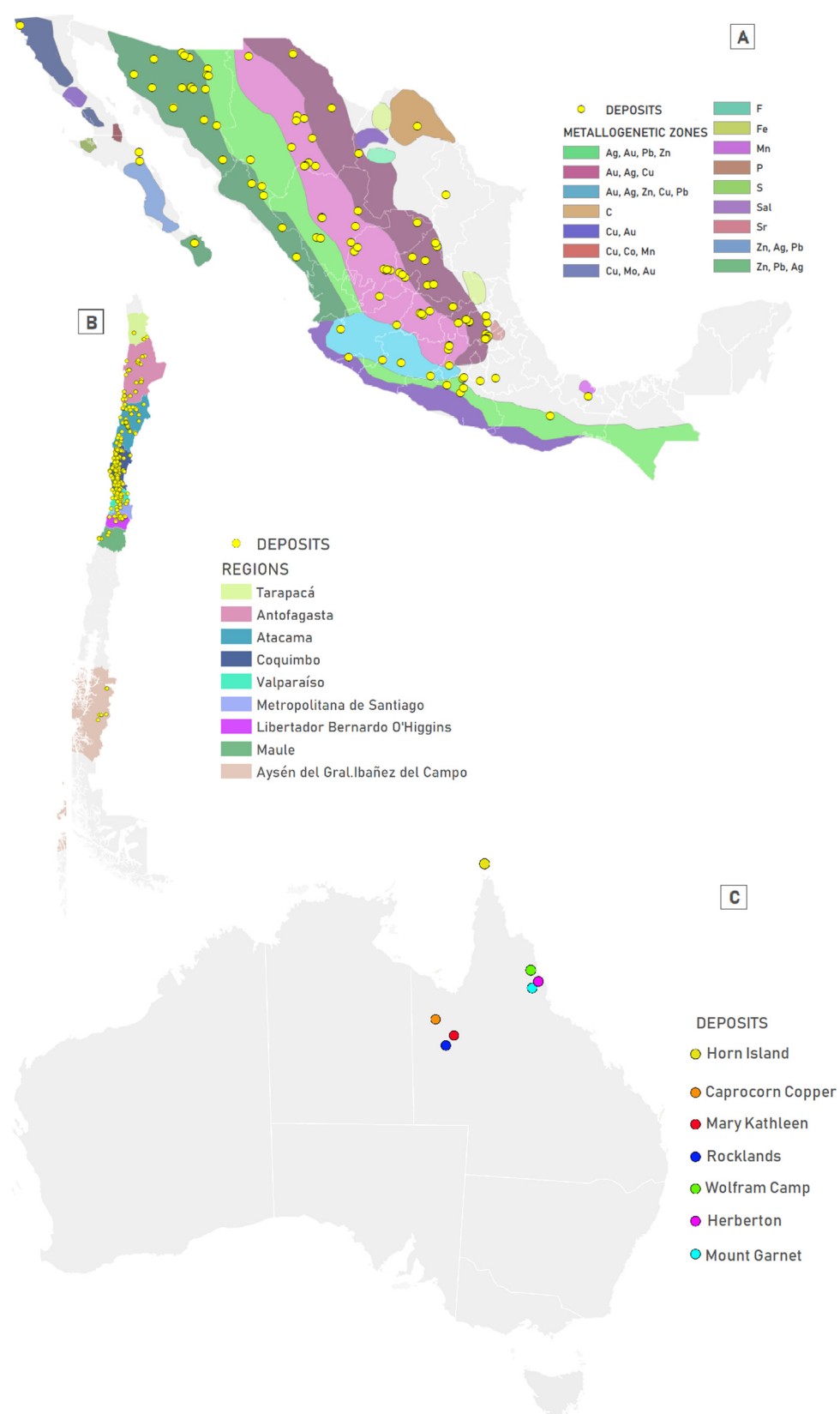

**Figure 2.** Spatial distribution of the data collected for (**A**) Mexico (alongside metallogenic province), (**B**) Chile (alongside regions) and (**C**) Australia.

**Table 2.** Summary of the total number of identified deposits and samples and average concentrations of elements classified as critical and with economic relevance and potential ratio. The potential ratio was calculated by dividing the average value of each element by crustal abundance (Table 1).

| Country | | Mexico | | Chile | | Australia | |
|---|---|---|---|---|---|---|---|
| Deposits-Samples | | 159 | 274 | 642 | 2032 | 7 | 490 |
| Critical Elements | | Average (ppm) | Potential Ratio | Average (ppm) | Potential Ratio | Average (ppm) | Potential Ratio |
| Platinum-group metals | Iridium (Ir) | ND | ND | ND | ND | ND | ND |
| | Palladium (Pd) | 1 | 66.7 | ND | ND | ND | ND |
| | Platinum (Pt) | ND | ND | ND | ND | ND | ND |
| | Rhodium (Rh) | ND | ND | ND | ND | ND | ND |
| | Ruthenium (Ru) | ND | ND | ND | ND | ND | ND |
| Rare earth elements | Praseodymium (Pr) | 9.3 | 1 | 4.13 | 0.45 | 158.11 | 17.19 |
| | Neodymium (Nd) | 14.2 | 0.3 | 16.62 | 0.40 | 370.36 | 8.92 |
| | Cerium (Ce) | 22.9 | 0.3 | 32.99 | 0.50 | 49.24 | 0.74 |
| | Terbium (Tb) | 3.4 | 2.8 | 0.45 | 0.38 | 1.37 | 1.14 |
| | Dysprosium (Dy) | 8.1 | 1.6 | 2.58 | 0.50 | 8.28 | 1.59 |
| | Yttrium (Y) | 12.2 | 0.4 | 41.14 | 1.25 | ND | ND |
| Other critical elements | Antimony (Sb) | 3169.6 | 15,847.8 | 91.47 | 457.35 | 31.12 | 155.58 |
| | Beryllium (Be) | 2.3 | 0.8 | ND | ND | 205.57 | 73.42 |
| | Bismuth (Bi) | 567.1 | 66,721.2 | 54.94 | 6463.14 | 78.05 | 9182.81 |
| | Cobalt (Co) | 101.8 | 4.1 | 35.93 | 1.44 | 103.07 | 4.12 |
| | Chromium (Cr) | 65.5 | 0.6 | 157.16 | 1.54 | 46.67 | 0.46 |
| | Gallium (Ga) | 5.4 | 0.3 | ND | ND | 488.02 | 25.69 |
| | Germanium (Ge) | 0.3 | 0.2 | ND | ND | 1.16 | 0.77 |
| | Hafnium (Hf) | 1.1 | 0.4 | 3.71 | 1.24 | 3.79 | 1.26 |
| | Indium (In) | 2 | 7.8 | ND | ND | 4.16 | 16.64 |
| | Lithium (Li) | 26 | 1.3 | ND | ND | 20.35 | 1.02 |
| | Magnesium (Mg) | ND | ND | ND | ND | 35,000.00 | 1.50 |
| | Niobium (Nb) | 2.6 | 0.1 | 12.53 | 0.63 | 25.19 | 1.26 |
| | Phosphorus (P) | 71.5 | 0.1 | ND | ND | 1,007.81 | 0.96 |
| | Scandium (Sc) | 5.6 | 0.3 | 18.67 | 0.85 | 10.43 | 0.47 |
| | Silicon (Si) | ND | ND | ND | ND | 235,000.00 | 0.83 |
| | Strontium (Sr) | 673 | 1.8 | 200.67 | 0.54 | 43.96 | 0.12 |
| | Tantalum (Ta) | 1 | 0.5 | 2.46 | 1.23 | 0.88 | 0.44 |
| | Titanium (Ti) | ND | ND | ND | ND | 0.23 | 0.00 |
| | Tungsten (W) | 38 | 30.4 | 47.72 | 38.18 | 85.33 | 68.26 |
| | Vanadium (V) | 60.605 | 0.5 | 142.51 | 1.19 | 76.18 | 0.63 |
| Raw materials with high relative economic importance | Iron (Fe) ore | ND | ND | ND | ND | ND | ND |
| | Aluminium (Al) | ND | ND | ND | ND | 43,000.00 | 0.52 |
| | Nickel (Ni) | 115.6 | 1.38 | 44.67 | 0.53 | 39.31 | 0.47 |
| | Zinc (Zn) | 6769 | 96.7 | 1449.53 | 20.71 | 995.80 | 14.23 |
| | Molybdenum (Mo) | 15.5 | 12.89 | 33.42 | 27.85 | 50.83 | 42.36 |
| | Lanthanum (La) | 21.2 | 0.54 | 19.37 | 0.50 | 24.08 | 0.62 |

The bulk geochemical results of the three countries have also been compared with the average crustal abundance of the element for assessing the potential ratio (Table 2). It can be observed that Mexico displayed average concentrations with at least one order of magnitude greater than crustal abundance for 14 elements, although Sb, Bi and Zn greatly exceed values with 158,418, 66,721 and 97 potential ratios, respectively. Chile displayed average concentrations above crustal abundance for Bi, Sb, W, Mo, Zn, Cr, Co, Y, Hf, Ta and Va. Meanwhile, Bi reported an average that is ~6000 times higher than crustal; Sb exceed it between ~450 times; W, Mo and Zn are 20–40 times greater; and the rest of the elements constituted only about 1.2–1.5. In the case of Australia, 16 elements displayed average concentrations with at least one order of magnitude greater than crustal abundance with Bi ~9000 times higher than crustal; Sb exceeding ~150 times, and Mo, W, Be, Sb and Ga at least

20 times. It is noted that in the case of the less abundant elements, this concentration may fluctuate in several orders of magnitude [42].

*3.5. Database Analysis for Mexico*

The geospatial distribution of the Mexican sites along with their respective metallogenic province (Figure 2) show that most of the data collected belongs to the Ag-Au-Pb-Zn metallogenic province with 99 sites evaluated, followed by the Zn-Ag-Pb province with 74 sites and Cu-Mo-Au with 46 sites. As many of the tailings have yet to be fully characterised, it is expected that according to the metallogenic provinces additional important elements may be present in significant concentrations in addition to the high concentrations of metal(oid)s historically generated and measured from the mine wastes [43]. Based on a knowledge of the host minerals and companion elements [44], Ge, Ga, In, Tl, Bi, Sb, Zn and Mo among other elements are likely to be found in those areas with some evidence presented in the following sections.

The bulk results of select elements grouped by metallogenic province for Mexico are shown on a Tukey plot in Figure 3. Metallogenic provinces Au, Ag and Cu; Cu Mo and Au; Cu and Au; Ag, Au, Pb and Zn; and Zn, Ag and Pb present critical elements. The Cu, Au region was identified for Sb and Zn at average values of 9550 ppm and 27,024 ppm, respectively, but also contain other elements, such as Cr, Sr, Ni and Co, in lower concentrations as compared to crustal abundance (Table 1). In the Zn, Ag and Pb region, the average concentrations include Be (4.2 ppm), Co (472 ppm), Cr (62.99 ppm), V (70.75 ppm) and Zn (847 ppm) as well as minor content of Ni (62.54 ppm). For the Ag, Au, Pb, Zn region, richness in critical elements could be found even when some of the values are below, and the crustal abundances of others were very high, for instance, Pr (50.43 ppm), Nd (39.73 ppm), Ce (55.87 ppm), Tb (18.51 ppm), Dy (14.13 ppm), Y (22.41 ppm), Be (2.85 ppm), Hf (2.91 ppm), Nb (5 ppm), Sc (8.74 ppm) and La (90.46 ppm), with presence but lower concentrations of Zn, Ni, Sr, Sb, Co, Cr and Mo (Figure 3).

Statistical analysis of the critical elements identified in each region indicated that Michoacán and Sonora are the states with the largest number of critical elements and raw materials with high relative economic importance (Table 3). In the case of Sonora, the state is located in Au, Ag, Cu and Cu, Mo and Au metallogenic provinces in the northwest of Mexico (Figure 2), where the potential recovery potential based on the potential ratio outstood too for Bi, Sb, W, In, Zn and Mo (Table 2). In this region, the average concentration in parts-per notation (ppm) of the elements ranges from 64 Mo, 4638 Sb, 1103Bi, 31,798Zn, 76 W to 1.76 in (Table 3).

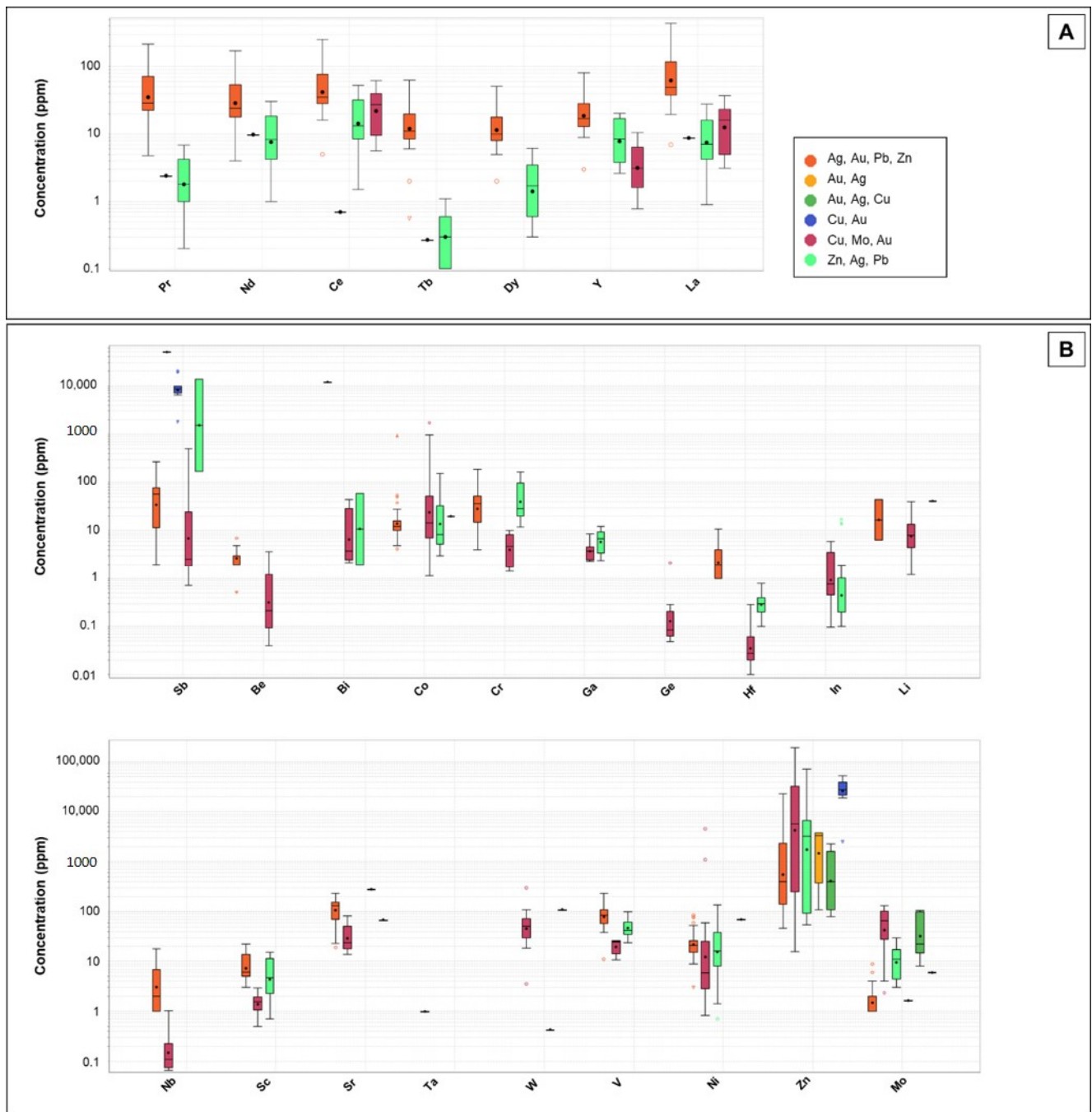

**Figure 3.** Tukey log-plot graph showing the concentration of the critical elements at each site included in Australia split in (**A**) REEs-Praseodymium, Neodymium, Cerium, Terbium, Dysprosium and Yttrium (**B**) other critical elements. The box is the middle 50% of data from Q1 to Q3. The outliers (circles) are in the top or bottom 5% of the data. The whiskers are the 5% and 95% values.

**Table 3.** Summary statistics of the elements classified as critical or with high economic relevance in each state from the Mexican data used in this study. Highlighted figures indicate a concentration above crustal abundance. PGE: Platinum Group Elements; SD: standard deviation; Min: minimum value; Max: maximum value.

| Element (ppm) | | PGE | Rare Earth Elements | | | | | |
|---|---|---|---|---|---|---|---|---|
| | | Pd | Pr | Nd | Ce | Tb | Dy | Y |
| Baja California | Min. | — | — | — | — | — | — | — |
| | Max. | — | — | — | — | — | — | — |
| | Mean | — | — | — | — | — | — | — |
| | SD | — | — | — | — | — | — | — |
| Baja California Sur | Min. | — | 0.0 | 0.1 | 0.3 | 0.0 | — | — |
| | Max. | — | 0.0 | 0.1 | 0.3 | 0.0 | — | — |
| | Mean | — | 0.0 | 0.1 | 0.3 | 0.0 | — | — |
| | SD | — | — | — | — | — | — | — |
| Chihuahua | Min. | — | — | — | — | — | — | — |
| | Max. | — | — | — | — | — | — | — |
| | Mean | — | — | — | — | — | — | — |
| | SD | — | — | — | — | — | — | — |
| Ciudad de México | Min. | — | — | — | — | — | — | — |
| | Max. | — | — | — | — | — | — | — |
| | Mean | — | — | — | — | — | — | — |
| | SD | — | — | — | — | — | — | — |
| Coahuila | Min. | — | — | — | — | — | — | — |
| | Max. | — | — | — | — | — | — | — |
| | Mean | — | — | — | — | — | — | — |
| | SD | — | — | — | — | — | — | — |
| Colima | Min. | — | — | — | — | — | — | — |
| | Max. | — | — | — | — | — | — | — |
| | Mean | — | — | — | — | — | — | — |
| | SD | — | — | — | — | — | — | — |
| Durango | Min. | — | — | — | — | — | — | — |
| | Max. | — | — | — | — | — | — | — |
| | Mean | — | — | — | — | — | — | — |
| | SD | — | — | — | — | — | — | — |
| Estado de México | Min. | — | — | — | — | — | — | — |
| | Max. | — | — | — | — | — | — | — |
| | Mean | — | — | — | — | — | — | — |
| | SD | — | — | — | — | — | — | — |
| Guanajuato | Min. | — | — | — | — | — | — | — |
| | Max. | — | — | — | — | — | — | — |
| | Mean | — | — | — | — | — | — | — |
| | SD | — | — | — | — | — | — | — |

**Table 3.** *Cont.*

| Element (ppm) | | PGE | Rare Earth Elements | | | | | |
|---|---|---|---|---|---|---|---|---|
| | | Pd | Pr | Nd | Ce | Tb | Dy | Y |
| Guerrero | Min. | — | 2.4 | 9.8 | 0.7 | 0.3 | — | — |
| | Max. | — | 2.4 | 9.8 | 0.7 | 0.3 | — | — |
| | Mean | — | 2.4 | 9.8 | 0.7 | 0.3 | — | — |
| | SD | — | — | — | — | — | — | — |
| Hidalgo | Min. | — | 2.7 | 10.8 | 10.0 | 0.3 | — | — |
| | Max. | — | 2.7 | 10.8 | 21.4 | 0.3 | — | — |
| | Mean | — | 2.7 | 10.8 | 15.7 | 0.3 | — | — |
| | SD | — | — | — | 8.0 | — | — | — |
| Jalisco | Min. | — | — | — | — | — | — | — |
| | Max. | — | — | — | — | — | — | — |
| | Mean | — | — | — | — | — | — | — |
| | SD | — | — | — | — | — | — | — |
| Michoacán | Min. | 1.0 | 4.1 | 4.0 | 5.0 | 2.0 | 2.0 | 3.0 |
| | Max. | 1.0 | 215.0 | 172.0 | 252.0 | 132.0 | 51.0 | 81.0 |
| | Mean | 1.0 | 50.4 | 39.7 | 55.9 | 18.5 | 14.1 | 22.4 |
| | SD | 0.0 | 48.7 | 37.4 | 51.5 | 20.9 | 10.6 | 17.0 |
| Morelos | Min. | — | — | — | — | — | — | — |
| | Max. | — | — | — | — | — | — | — |
| | Mean | — | — | — | — | — | — | — |
| | SD | — | — | — | — | — | — | — |
| Nuevo León | Min. | — | — | — | — | — | — | — |
| | Max. | — | — | — | — | — | — | — |
| | Mean | — | — | — | — | — | — | — |
| | SD | — | — | — | — | — | — | — |
| Oaxaca | Min. | — | — | — | — | — | — | — |
| | Max. | — | — | — | — | — | — | — |
| | Mean | — | — | — | — | — | — | — |
| | SD | — | — | — | — | — | — | — |
| Querétaro | Min. | — | — | — | — | — | — | — |
| | Max. | — | — | — | — | — | — | — |
| | Mean | — | — | — | — | — | — | — |
| | SD | — | — | — | — | — | — | — |
| San Luis Potosí | Min. | — | 2.3 | 8.3 | 19.3 | 0.3 | — | — |
| | Max. | — | 2.3 | 8.3 | 19.3 | 0.3 | — | — |
| | Mean | — | 2.3 | 8.3 | 19.3 | 0.3 | — | — |
| | SD | — | — | — | — | — | — | — |
| Sinaloa | Min. | — | — | — | — | — | — | — |
| | Max. | — | — | — | — | — | — | — |
| | Mean | — | — | — | — | — | — | — |
| | SD | — | — | — | — | — | — | — |

**Table 3.** *Cont.*

| Element (ppm) | | PGE | Rare Earth Elements | | | | | |
|---|---|---|---|---|---|---|---|---|
| | | Pd | Pr | Nd | Ce | Tb | Dy | Y |
| Sonora | Min. | — | — | — | 5.6 | — | — | 0.8 |
| | Max. | — | — | — | 62.2 | — | — | 10.5 |
| | Mean | — | — | — | 28.6 | — | — | 4.2 |
| | SD | — | — | — | 18.7 | — | — | 3.3 |
| Veracruz | Min. | — | — | — | — | — | — | — |
| | Max. | — | — | — | — | — | — | — |
| | Mean | — | — | — | — | — | — | — |
| | SD | — | — | — | — | — | — | — |
| Zacatecas | Min. | — | 4.8 | 19.3 | 42.0 | 0.6 | — | — |
| | Max. | — | 4.8 | 19.3 | 42.0 | 0.6 | — | — |
| | Mean | — | 4.8 | 19.3 | 42.0 | 0.6 | — | — |
| | SD | — | — | — | — | — | — | — |

| Element (ppm) | | Others Elements with Critical and Economic Importance | | | | | | | | | |
|---|---|---|---|---|---|---|---|---|---|---|---|
| | | Sb | Be | Bi | Co | Cr | Ga | Ge | Hf | In | Li |
| Baja California | Min. | — | — | — | — | — | — | — | — | — | — |
| | Max. | — | — | — | — | — | — | — | — | — | — |
| | Mean | — | — | — | — | — | — | — | — | — | — |
| | SD | — | — | — | — | — | — | — | — | — | — |
| Baja California Sur | Min. | 1,780.0 | — | — | 54.0 | 222.0 | — | — | — | — | — |
| | Max. | 20,200.0 | — | — | 54.0 | 222.0 | — | — | — | — | — |
| | Mean | 9,550.0 | — | — | 54.0 | 222.0 | — | — | — | — | — |
| | SD | 5,463.7 | — | — | — | — | — | — | — | — | — |
| Chihuahua | Min. | 270.0 | — | — | — | 165.5 | — | — | — | — | — |
| | Max. | 270.0 | — | — | — | 165.5 | — | — | — | — | — |
| | Mean | 270.0 | — | — | — | 165.5 | — | — | — | — | — |
| | SD | — | — | — | — | — | — | — | — | — | — |
| Ciudad de México | Min. | — | — | — | — | — | — | — | — | — | — |
| | Max. | — | — | — | — | — | — | — | — | — | — |
| | Mean | — | — | — | — | — | — | — | — | — | — |
| | SD | — | — | — | — | — | — | — | — | — | — |
| Coahuila | Min. | — | — | — | — | — | — | — | — | — | — |
| | Max. | — | — | — | — | — | — | — | — | — | — |
| | Mean | — | — | — | — | — | — | — | — | — | — |
| | SD | — | — | — | — | — | — | — | — | — | — |
| Colima | Min. | — | — | — | — | — | — | — | — | — | — |
| | Max. | — | — | — | — | — | — | — | — | — | — |
| | Mean | — | — | — | — | — | — | — | — | — | — |
| | SD | — | — | — | — | — | — | — | — | — | — |

**Table 3.** *Cont.*

| Element (ppm) | | Others Elements with Critical and Economic Importance | | | | | | | | | |
|---|---|---|---|---|---|---|---|---|---|---|---|
| | | Sb | Be | Bi | Co | Cr | Ga | Ge | Hf | In | Li |
| Durango | Min. | — | — | — | — | 6.0 | — | — | — | — | — |
| | Max. | — | — | — | — | 24.8 | — | — | — | — | — |
| | Mean | — | — | — | — | 16.1 | — | — | — | — | — |
| | SD | — | — | — | — | 8.3 | — | — | — | — | — |
| Estado de México | Min. | — | — | — | — | — | — | — | — | — | — |
| | Max. | — | — | — | — | — | — | — | — | — | — |
| | Mean | — | — | — | — | — | — | — | — | — | — |
| | SD | — | — | — | — | — | — | — | — | — | — |
| Guanajuato | Min. | 7.8 | 3.6 | — | 7.6 | 6.1 | — | — | — | — | — |
| | Max. | 7.8 | 5.0 | — | 937.0 | 174.0 | — | — | — | — | — |
| | Mean | 7.8 | 4.2 | — | 472.3 | 63.0 | — | — | — | — | — |
| | SD | — | 0.7 | — | 657.2 | 52.8 | — | — | — | — | — |
| Guerrero | Min. | — | — | — | — | — | — | — | — | — | — |
| | Max. | — | — | — | — | — | — | — | — | — | — |
| | Mean | — | — | — | — | — | — | — | — | — | — |
| | SD | — | — | — | — | — | — | — | — | — | — |
| Hidalgo | Min. | 170.0 | — | 2.0 | 3.0 | 19.0 | — | — | — | — | — |
| | Max. | 170.0 | — | 60.0 | 1710.0 | 141.0 | — | — | — | — | — |
| | Mean | 170.0 | — | 31.0 | 224.1 | 49.4 | — | — | — | — | — |
| | SD | — | — | 41.0 | 497.2 | 43.4 | — | — | — | — | — |
| Jalisco | Min. | — | — | — | — | — | — | — | — | — | — |
| | Max. | — | — | — | — | — | — | — | — | — | — |
| | Mean | — | — | — | — | — | — | — | — | — | — |
| | SD | — | — | — | — | — | — | — | — | — | — |
| Michoacán | Min. | 2.0 | 2.0 | — | 5.0 | 4.0 | — | — | 1.0 | — | — |
| | Max. | 133.0 | 7.0 | — | 54.0 | 91.0 | — | — | 11.0 | — | — |
| | Mean | 55.9 | 2.9 | — | 15.5 | 35.0 | — | — | 2.9 | — | — |
| | SD | 40.0 | 1.0 | — | 9.8 | 23.6 | — | — | 2.5 | — | — |
| Morelos | Min. | — | — | — | 20.0 | — | — | — | — | — | 41.3 |
| | Max. | — | — | — | 20.0 | — | — | — | — | — | 41.3 |
| | Mean | — | — | — | 20.0 | — | — | — | — | — | 41.3 |
| | SD | — | — | — | — | — | — | — | — | — | — |
| Nuevo León | Min. | 58.9 | — | — | — | — | — | — | — | — | — |
| | Max. | 58.9 | — | — | — | — | — | — | — | — | — |
| | Mean | 58.9 | — | — | — | — | — | — | — | — | — |
| | SD | — | — | — | — | — | — | — | — | — | — |
| Oaxaca | Min. | — | — | — | — | — | — | — | — | — | — |
| | Max. | — | — | — | — | — | — | — | — | — | — |
| | Mean | — | — | — | — | — | — | — | — | — | — |
| | SD | — | — | — | — | — | — | — | — | — | — |

**Table 3.** *Cont.*

| Element (ppm) | | Others Elements with Critical and Economic Importance | | | | | | | | | |
|---|---|---|---|---|---|---|---|---|---|---|---|
| | | Sb | Be | Bi | Co | Cr | Ga | Ge | Hf | In | Li |
| Querétaro | Min. | — | — | — | — | — | — | — | — | — | — |
| | Max. | — | — | — | — | — | — | — | — | — | — |
| | Mean | — | — | — | — | — | — | — | — | — | — |
| | SD | — | — | — | — | — | — | — | — | — | — |
| San Luis Potosí | Min. | 13,743.0 | — | — | 4.3 | 11.8 | — | — | — | — | — |
| | Max. | 13,743.0 | — | — | 4.3 | 11.8 | — | — | — | — | — |
| | Mean | 13,743.0 | — | — | 4.3 | 11.8 | — | — | — | — | — |
| | SD | — | — | — | — | — | — | — | — | — | — |
| Sinaloa | Min. | — | — | — | — | — | — | — | — | — | — |
| | Max. | — | — | — | — | — | — | — | — | — | — |
| | Mean | — | — | — | — | — | — | — | — | — | — |
| | SD | — | — | — | — | — | — | — | — | — | — |
| Sonora | Min. | 0.7 | 0.0 | 2.2 | 1.1 | 1.4 | 2.3 | 0.0 | 0.0 | 0.1 | 1.2 |
| | Max. | 50,000.0 | 3.7 | 12,000.0 | 51.3 | 10.1 | 8.7 | 2.2 | 0.3 | 6.1 | 39.9 |
| | Mean | 4638.2 | 0.8 | 1103.3 | 16.3 | 5.2 | 4.1 | 0.3 | 0.1 | 1.8 | 11.5 |
| | SD | 15,046.1 | 1.1 | 3614.1 | 15.9 | 3.4 | 1.9 | 0.6 | 0.1 | 2.0 | 11.5 |
| Veracruz | Min. | — | — | — | — | 4.2 | — | — | — | — | — |
| | Max. | — | — | — | — | 4.2 | — | — | — | — | — |
| | Mean | — | — | — | — | 4.2 | — | — | — | — | — |
| | SD | — | — | — | — | — | — | — | — | — | — |
| Zacatecas | Min. | 4.1 | 0.5 | — | 4.2 | 8.8 | — | — | — | — | 6.4 |
| | Max. | 60.3 | 2.6 | — | 11.0 | 189.0 | — | — | — | — | 44.1 |
| | Mean | 32.2 | 1.5 | — | 7.6 | 82.7 | — | — | — | — | 25.3 |
| | SD | 39.7 | 1.4 | — | 4.8 | 94.4 | — | — | — | — | 26.6 |

| Element (ppm) | | Others Elements with Critical and Economic Importance | | | | | | | | | |
|---|---|---|---|---|---|---|---|---|---|---|---|
| | | Nb | Sc | Sr | Ta | W | V | Ni | Zn | Mo | La |
| Baja California | Min. | — | — | — | — | — | — | — | — | 6.0 | — |
| | Max. | — | — | — | — | — | — | — | — | 6.0 | — |
| | Mean | — | — | — | — | — | — | — | — | 6.0 | — |
| | SD | — | — | — | — | — | — | — | — | — | — |
| Baja California Sur | Min. | — | — | 2253.0 | — | — | — | 627.0 | 660.0 | — | 0.2 |
| | Max. | — | — | 2253.0 | — | — | — | 627.0 | 52,200.0 | — | 0.2 |
| | Mean | — | — | 2253.0 | — | — | — | 627.0 | 27,024.4 | — | 0.2 |
| | SD | — | — | — | — | — | — | — | 13,785.0 | — | — |
| Chihuahua | Min. | — | — | — | — | — | — | — | 520.0 | 8.0 | — |
| | Max. | — | — | — | — | — | — | — | 12,531.0 | 8.0 | — |
| | Mean | — | — | — | — | — | — | — | 5829.6 | 8.0 | — |
| | SD | — | — | — | — | — | — | — | 3908.8 | — | — |
| Ciudad de México | Min. | — | — | — | — | — | — | — | 1540.0 | — | — |
| | Max. | — | — | — | — | — | — | — | 1540.0 | — | — |
| | Mean | — | — | — | — | — | — | — | 1540.0 | — | — |
| | SD | — | — | — | — | — | — | — | — | — | — |

**Table 3.** *Cont.*

| Element (ppm) | | Others Elements with Critical and Economic Importance | | | | | | | | | |
|---|---|---|---|---|---|---|---|---|---|---|---|
| | | Nb | Sc | Sr | Ta | W | V | Ni | Zn | Mo | La |
| Coahuila | Min. | — | — | — | — | — | — | — | 3600.0 | — | — |
| | Max. | — | — | — | — | — | — | — | 3600.0 | — | — |
| | Mean | — | — | — | — | — | — | — | 3600.0 | — | — |
| | SD | — | — | — | — | — | — | — | — | — | — |
| Colima | Min. | — | — | — | — | — | — | — | — | — | — |
| | Max. | — | — | — | — | — | — | — | — | — | — |
| | Mean | — | — | — | — | — | — | — | — | — | — |
| | SD | — | — | — | — | — | — | — | — | — | — |
| Durango | Min. | — | — | — | — | — | — | — | 300.0 | — | — |
| | Max. | — | — | — | — | — | — | — | 23,000.0 | — | — |
| | Mean | — | — | — | — | — | — | — | 7365.0 | — | — |
| | SD | — | — | — | — | — | — | — | 10,524.7 | — | — |
| Estado de México | Min. | — | — | — | — | — | — | — | 5700.0 | — | — |
| | Max. | — | — | — | — | — | — | — | 5700.0 | — | — |
| | Mean | — | — | — | — | — | — | — | 5700.0 | — | — |
| | SD | — | — | — | — | — | — | — | — | — | — |
| Guanajuato | Min. | — | — | — | — | — | 48.0 | 38.7 | 47.0 | — | — |
| | Max. | — | — | — | — | — | 127.0 | 84.0 | 2811.0 | — | — |
| | Mean | — | — | — | — | — | 70.8 | 62.5 | 846.6 | — | — |
| | SD | — | — | — | — | — | 37.8 | 18.5 | 1145.8 | — | — |
| Guerrero | Min. | — | — | — | — | — | — | — | 1305.1 | — | 8.8 |
| | Max. | — | — | — | — | — | — | — | 3735.0 | — | 8.8 |
| | Mean | — | — | — | — | — | — | — | 2995.0 | — | 8.8 |
| | SD | — | — | — | — | — | — | — | 1135.9 | — | — |
| Hidalgo | Min. | — | — | — | — | — | 24.0 | 6.0 | 1573.0 | 3.0 | 12.2 |
| | Max. | — | — | — | — | — | 100.0 | 4550.0 | 12,413.0 | 18.0 | 12.2 |
| | Mean | — | — | — | — | — | 52.1 | 283.0 | 5887.3 | 10.0 | 12.2 |
| | SD | — | — | — | — | — | 24.4 | 957.2 | 3,492.4 | 6.3 | — |
| Jalisco | Min. | — | — | — | — | — | — | — | 3,370.0 | — | — |
| | Max. | — | — | — | — | — | — | — | 3370.0 | — | — |
| | Mean | — | — | — | — | — | — | — | 3370.0 | — | — |
| | SD | — | — | — | — | — | — | — | — | — | — |
| Michoacán | Min. | 1.0 | 3.0 | 19.0 | 1.0 | — | 11.0 | 3.0 | 64.0 | 1.0 | 6.4 |
| | Max. | 18.0 | 22.0 | 236.0 | 1.0 | — | 233.0 | 77.0 | 2630.0 | 6.0 | 442.0 |
| | Mean | 5.0 | 8.7 | 121.8 | 1.0 | — | 89.8 | 22.1 | 287.2 | 1.7 | 90.5 |
| | SD | 5.3 | 5.4 | 53.5 | 0.0 | — | 43.6 | 11.8 | 372.0 | 1.1 | 93.2 |
| Morelos | Min. | — | — | 280.0 | — | 0.4 | — | — | 109.0 | 1.7 | — |
| | Max. | — | — | 280.0 | — | 0.4 | — | — | 109.0 | 1.7 | — |
| | Mean | — | — | 280.0 | — | 0.4 | — | — | 109.0 | 1.7 | — |
| | SD | — | — | — | — | — | — | — | — | — | — |
| Nuevo León | Min. | — | — | — | — | — | — | 31.0 | 16,897.0 | — | — |
| | Max. | — | — | — | — | — | — | 31.0 | 16,897.0 | — | — |
| | Mean | — | — | — | — | — | — | 31.0 | 16,897.0 | — | — |
| | SD | — | — | — | — | — | — | — | — | — | — |
| Oaxaca | Min. | — | — | — | — | — | — | — | 3800.0 | — | — |
| | Max. | — | — | — | — | — | — | — | 3800.0 | — | — |
| | Mean | — | — | — | — | — | — | — | 3800.0 | — | — |
| | SD | — | — | — | — | — | — | — | — | — | — |

**Table 3.** *Cont.*

| Element (ppm) | | Others Elements with Critical and Economic Importance | | | | | | | | | |
|---|---|---|---|---|---|---|---|---|---|---|---|
| | | Nb | Sc | Sr | Ta | W | V | Ni | Zn | Mo | La |
| Querétaro | Min. | — | — | — | — | — | — | — | 996.0 | 30.0 | — |
| | Max. | — | — | — | — | — | — | — | 8786.0 | 30.0 | — |
| | Mean | — | — | — | — | — | — | — | 3951.4 | 30.0 | — |
| | SD | — | — | — | — | — | — | — | 3014.6 | — | — |
| San Luis Potosí | Min. | — | — | — | — | — | 41.8 | 16.4 | 65.1 | 12.4 | 11.0 |
| | Max. | — | — | — | — | — | 41.8 | 128.8 | 5240.0 | 12.4 | 11.0 |
| | Mean | — | — | — | — | — | 41.8 | 72.6 | 3216.3 | 12.4 | 11.0 |
| | SD | — | — | — | — | — | — | 79.5 | 2220.1 | — | — |
| Sinaloa | Min. | — | — | — | — | — | — | — | 180.0 | 15.0 | — |
| | Max. | — | — | — | — | — | — | — | 1230.0 | 15.0 | — |
| | Mean | — | — | — | — | — | — | — | 705.0 | 15.0 | — |
| | SD | — | — | — | — | — | — | — | 742.5 | — | — |
| Sonora | Min. | 0.1 | 0.5 | 14.1 | — | 3.5 | 10.9 | 0.8 | 15.7 | 2.3 | 3.1 |
| | Max. | 1.0 | 2.9 | 80.6 | — | 300.0 | 26.4 | 69.5 | 192,000. | 133.0 | 37.4 |
| | Mean | 0.3 | 1.5 | 37.3 | — | 75.5 | 20.5 | 10.8 | 31,798.4 | 63.9 | 16.6 |
| | SD | 0.4 | 0.7 | 23.0 | — | 77.6 | 6.2 | 19.7 | 49,013.3 | 40.9 | 11.2 |
| Veracruz | Min. | — | — | — | — | — | — | 9.3 | 5.3 | — | — |
| | Max. | — | — | — | — | — | — | 9.3 | 5.3 | — | — |
| | Mean | — | — | — | — | — | — | 9.3 | 5.3 | — | — |
| | SD | — | — | — | — | — | — | — | — | — | — |
| Zacatecas | Min. | — | — | — | — | — | 59.1 | 19.1 | 1170.0 | 3.1 | 19.4 |
| | Max. | — | — | — | — | — | 118.2 | 22.1 | 8200.0 | 8.9 | 19.4 |
| | Mean | — | — | — | — | — | 88.6 | 20.2 | 3230.6 | 6.0 | 19.4 |
| | SD | — | — | — | — | — | 41.8 | 1.6 | 2485.0 | 4.1 | — |

### 3.6. Database Analysis for Chile

The extensive database for Chile allowed for a broad analysis of this information in this study (Figure 4 and Table 4). The geochemical results of select elements grouped by region for Chile show that for northern and central Chile, Zn values are the highest for all regions with over 500 ppm, which is seven times more than the crustal abundance, with the highest average for the Antofagasta Region with 3747 ppm and a maximum of 30,199 ppm. These regions present richness also in the following critical and economic elements: Co, Mo, W, Sb, Bi, Sc, V, Cr and REE (Y). Co shows a maximum mean concentration in the Valparaiso region (average: 93 ppm; maximum: 4175 ppm). For Mo, the highest mean of 143 ppm was measured in Antofagasta with a maximum of 1403 ppm. W shows the highest mean for Atacama and Coquimbo regions, 53 ppm and 49 ppm, respectively. Sb and Bi reveal a maximum average concentration in the Antofagasta Region (631 ppm and 186 ppm respectively), which is 3000 and 21,500 times higher than crustal abundance.

**Table 4.** Summary statistics of the elements classified as critical or with high economic relevance in each region from the Chilean data used in this study. Highlighted figures indicate a concentration above crustal abundance. SD: standard deviation; Min: minimum value; Max: maximum value.

| Element (ppm) | | | Rare Earth Elements | | | | | |
|---|---|---|---|---|---|---|---|---|
| | | | Pr | Nd | Ce | Tb | Dy | Y |
| North | Tarapacá | Min. | 0.5 | 1.9 | 3.8 | 0.1 | 0.3 | 12.0 |
| | | Max. | 5.8 | 18.7 | 55.4 | 0.7 | 2.3 | 78.0 |
| | | Mean | 2.4 | 8.7 | 20.2 | 0.2 | 1.2 | 27.6 |
| | | SD | 1.8 | 6.2 | 17.3 | 0.2 | 0.7 | 28.3 |

Table 4. *Cont.*

| Element (ppm) | | | Rare Earth Elements | | | | | |
|---|---|---|---|---|---|---|---|---|
| | | | **Pr** | **Nd** | **Ce** | **Tb** | **Dy** | **Y** |
| | Antofagasta | Min. | 0.3 | 1.0 | 2.9 | 0.0 | 0.2 | 10.0 |
| | | Max. | 50.0 | 148.1 | 744.2 | 1.2 | 7.0 | 83.0 |
| | | Mean | 5.2 | 20.2 | 46.6 | 0.4 | 2.5 | 39.2 |
| | | SD | 5.6 | 16.9 | 81.2 | 0.2 | 1.4 | 18.0 |
| | Atacama | Min. | 0.4 | 1.5 | 3.1 | 0.0 | 0.2 | 10.0 |
| | | Max. | 24.7 | 70.4 | 273.0 | 4.9 | 32.4 | 4245.0 |
| | | Mean | 3.8 | 14.8 | 31.7 | 0.4 | 2.4 | 73.0 |
| | | SD | 2.7 | 9.7 | 25.7 | 0.4 | 2.2 | 329.9 |
| | Coquimbo | Min. | 0.5 | 1.8 | 4.3 | 0.1 | 0.3 | 10.0 |
| | | Max. | 44.0 | 164.7 | 331.6 | 3.2 | 16.5 | 101.0 |
| | | Mean | 4.2 | 17.1 | 32.9 | 0.5 | 2.7 | 33.9 |
| | | SD | 2.8 | 11.0 | 23.3 | 0.3 | 1.4 | 15.5 |
| Central | Valparaíso | Min. | 0.4 | 1.5 | 3.0 | 0.0 | 0.2 | 10.0 |
| | | Max. | 8.8 | 34.3 | 72.9 | 1.2 | 7.0 | 90.0 |
| | | Mean | 3.3 | 13.6 | 25.7 | 0.4 | 2.3 | 27.8 |
| | | SD | 1.7 | 6.9 | 13.1 | 0.2 | 1.3 | 13.8 |
| | Metropolitana | Min. | 0.6 | 2.2 | 4.5 | 0.1 | 0.3 | 11.0 |
| | | Max. | 6.4 | 27.6 | 51.1 | 0.9 | 5.1 | 55.0 |
| | | Mean | 2.7 | 11.2 | 21.5 | 0.3 | 1.7 | 30.5 |
| | | SD | 1.6 | 6.6 | 11.5 | 0.2 | 1.1 | 13.0 |
| | O'Higgins | Min. | 1.3 | 5.0 | 11.1 | 0.1 | 0.6 | 16.0 |
| | | Max. | 15.2 | 56.4 | 111.3 | 2.6 | 18.9 | 72.0 |
| | | Mean | 3.5 | 14.4 | 27.8 | 0.4 | 2.2 | 41.7 |
| | | SD | 2.3 | 8.4 | 17.1 | 0.4 | 3.1 | 15.5 |
| | Maule | Min. | 4.8 | 16.0 | 34.9 | 0.3 | 2.0 | 48.0 |
| | | Max. | 6.1 | 20.5 | 75.7 | 0.6 | 3.6 | 59.0 |
| | | Mean | 5.4 | 18.1 | 57.0 | 0.5 | 2.8 | 52.0 |
| | | SD | 0.7 | 2.3 | 20.6 | 0.1 | 0.8 | 6.1 |
| South | Aysén | Min. | 0.8 | 2.5 | 8.1 | 0.0 | 0.2 | 11.0 |
| | | Max. | 64.5 | 263.1 | 517.5 | 6.0 | 33.5 | 71.0 |
| | | Mean | 13.1 | 52.2 | 100.7 | 1.4 | 7.3 | 28.8 |
| | | SD | 18.3 | 73.8 | 139.7 | 1.9 | 9.4 | 19.6 |

| Element (ppm) | | | Others Elements with Critical and Economic Importance | | | | | | | | | | | | | | |
|---|---|---|---|---|---|---|---|---|---|---|---|---|---|---|---|---|---|
| | | | **Sb** | **Bi** | **Co** | **Cr** | **Hf** | **Nb** | **Sc** | **Sr** | **Ta** | **W** | **V** | **Ni** | **Zn** | **Mo** | **La** |
| North | Tarapacá | Min. | 15.9 | 14.9 | 8.0 | 36.0 | 0.2 | 5.0 | 13.0 | 39.0 | 0.2 | 96.2 | 27.0 | 6.0 | 69.0 | 5.2 | 2.0 |
| | | Max. | 671.2 | 51.8 | 76.0 | 61.0 | 5.9 | 32.0 | 26.0 | 761.0 | 1.7 | 96.2 | 193.0 | 78.0 | 7853.0 | 119.8 | 21.4 |
| | | Mean | 189.9 | 27.6 | 53.1 | 50.0 | 2.3 | 11.2 | 18.3 | 211.6 | 0.7 | 96.2 | 106.4 | 26.0 | 2419.2 | 37.6 | 7.7 |
| | | SD | 236.8 | 21.0 | 22.5 | 10.2 | 1.8 | 11.7 | 3.9 | 227.5 | 0.6 | NS | 51.9 | 34.8 | 2735.5 | 52.0 | 6.7 |
| | Antofagasta | Min. | 10.0 | 12.0 | 6.0 | 19.0 | 0.6 | 5.0 | 6.0 | 9.0 | 0.0 | 10.1 | 19.0 | 5.0 | 6.0 | 5.1 | 1.7 |
| | | Max. | 7777.8 | 359.2 | 113.0 | 677.0 | 10.5 | 58.0 | 39.0 | 1134.0 | 133.0 | 92.0 | 348.0 | 146.0 | 30,199 | 1403.2 | 665.3 |
| | | Mean | 631.4 | 185.6 | 30.6 | 92.1 | 3.9 | 13.4 | 19.9 | 340.2 | 2.5 | 32.0 | 107.8 | 43.5 | 3746.7 | 142.6 | 27.6 |
| | | SD | 1342.4 | 245.5 | 20.9 | 120.9 | 1.8 | 10.6 | 6.4 | 259.9 | 14.6 | 22.4 | 77.5 | 38.7 | 7288.0 | 271.5 | 70.9 |
| | Atacama | Min. | 10.2 | 10.0 | 5.0 | 11.0 | 0.6 | 5.0 | 5.0 | 5.0 | 0.0 | 10.0 | 16.0 | 5.0 | 5.0 | 5.0 | 0.2 |
| | | Max. | 6136.3 | 648.9 | 871.0 | 128,95 | 10.5 | 45.0 | 44.0 | 968.0 | 42.9 | 506.8 | 411.0 | 392.0 | 38,413 | 914.6 | 169.6 |
| | | Mean | 102.4 | 85.1 | 50.2 | 418.3 | 3.3 | 12.2 | 17.4 | 182.7 | 1.3 | 52.8 | 116.8 | 39.5 | 765.2 | 36.0 | 16.5 |
| | | SD | 370.0 | 118.1 | 92.3 | 6,098.1 | 1.8 | 8.0 | 6.2 | 153.9 | 3.9 | 73.9 | 52.0 | 38.1 | 2750.0 | 85.2 | 15.9 |
| | Coquimbo | Min. | 10.2 | 10.1 | 5.0 | 10.0 | 0.4 | 5.0 | 5.0 | 5.0 | 0.0 | 10.1 | 28.0 | 5.0 | 5.0 | 5.0 | 0.6 |
| | | Max. | 2778.7 | 179.9 | 765.0 | 1270.0 | 14.0 | 55.0 | 42.0 | 2386.0 | 144.7 | 1937.8 | 1084.0 | 158.0 | 32,834 | 412.7 | 689.2 |
| | | Mean | 57.6 | 28.3 | 23.3 | 72.4 | 3.7 | 12.8 | 18.9 | 206.7 | 2.3 | 49.1 | 161.5 | 50.1 | 534.8 | 19.7 | 21.0 |
| | | SD | 143.5 | 30.9 | 35.9 | 74.1 | 1.9 | 8.3 | 6.9 | 214.3 | 9.2 | 142.3 | 81.6 | 31.0 | 1736.1 | 27.8 | 40.4 |

Table 4. *Cont.*

| Element (ppm) | | | Others Elements with Critical and Economic Importance | | | | | | | | | | | | | | |
|---|---|---|---|---|---|---|---|---|---|---|---|---|---|---|---|---|---|
| | | | Sb | Bi | Co | Cr | Hf | Nb | Sc | Sr | Ta | W | V | Ni | Zn | Mo | La |
| Central | Valparaíso | Min. | 10.0 | 11.0 | 5.0 | 13.0 | 0.6 | 5.0 | 5.0 | 26.0 | 0.2 | 12.6 | 32.0 | 5.0 | 5.0 | 5.5 | 0.6 |
| | | Max. | 470.2 | 52.0 | 4175.0 | 1685.0 | 14.1 | 44.0 | 38.0 | 717.0 | 21.0 | 81.2 | 423.0 | 127.0 | 93,346 | 610.0 | 104.2 |
| | | Mean | 48.8 | 32.9 | 93.3 | 84.2 | 3.3 | 10.9 | 18.7 | 192.2 | 1.5 | 40.5 | 127.9 | 27.4 | 1330.6 | 59.1 | 11.9 |
| | | SD | 68.6 | 11.8 | 416.2 | 147.1 | 1.8 | 7.7 | 6.3 | 115.7 | 2.3 | 24.2 | 50.9 | 26.9 | 6773.5 | 116.2 | 8.8 |
| | Metropolitana | Min. | 10.8 | 10.3 | 7.0 | 15.0 | 0.5 | 5.0 | 6.0 | 8.0 | 0.2 | 11.0 | 22.0 | 6.0 | 26.0 | 5.1 | 0.9 |
| | | Max. | 302.0 | 50.0 | 57.0 | 5691.0 | 10.5 | 27.0 | 38.0 | 355.0 | 3.9 | 134.7 | 254.0 | 111.0 | 19,757 | 319.9 | 24.6 |
| | | Mean | 35.6 | 33.2 | 27.5 | 219.5 | 3.6 | 11.2 | 18.5 | 137.2 | 1.1 | 55.0 | 109.9 | 37.2 | 2051.0 | 34.7 | 10.0 |
| | | SD | 44.5 | 19.9 | 15.8 | 799.5 | 2.0 | 6.2 | 7.3 | 93.0 | 0.8 | 33.6 | 51.9 | 31.0 | 4836.3 | 57.4 | 5.8 |
| | O'Higgins | Min. | 10.7 | 40.1 | 9.0 | 39.0 | 1.5 | 6.0 | 6.0 | 13.0 | 0.3 | 19.2 | 36.0 | 6.0 | 62.0 | 7.7 | 3.9 |
| | | Max. | 58.4 | 87.0 | 17.0 | 157.0 | 7.6 | 38.0 | 39.0 | 329.0 | 19.5 | 50.5 | 266.0 | 155.0 | 8795.0 | 270.3 | 55.3 |
| | | Mean | 26.8 | 64.0 | 11.9 | 107.8 | 3.4 | 16.6 | 23.8 | 186.1 | 3.5 | 31.5 | 179.4 | 61.3 | 1144.9 | 102.8 | 14.7 |
| | | SD | 13.3 | 12.8 | 2.1 | 34.6 | 1.2 | 8.7 | 7.6 | 84.2 | 4.2 | 10.1 | 56.3 | 35.5 | 2179.7 | 61.8 | 10.2 |
| | Maule | Min. | 14.4 | 14.4 | 12.0 | 38.0 | 5.5 | 19.0 | 12.0 | 21.0 | 0.2 | — | 115.0 | 73.0 | 120.0 | 5.2 | 16.4 |
| | | Max. | 27.1 | 14.4 | 19.0 | 91.0 | 7.0 | 21.0 | 44.0 | 47.0 | 0.2 | — | 286.0 | 84.0 | 442.0 | 6.1 | 21.0 |
| | | Mean | 21.6 | 14.4 | 16.7 | 71.3 | 6.4 | 20.0 | 33.0 | 29.7 | 0.2 | — | 224.7 | 76.7 | 251.0 | 5.6 | 18.7 |
| | | SD | 6.5 | NS | 4.0 | 29.0 | 0.8 | 1.0 | 18.2 | 15.0 | NS | — | 95.2 | 6.4 | 169.2 | 0.5 | 2.3 |
| South | Aysén | Min. | 10.1 | 14.1 | 5.0 | 30.0 | 0.3 | 5.0 | 6.0 | 11.0 | 0.2 | — | 12.0 | 6.0 | 177.0 | 6.5 | 0.0 |
| | | Max. | 329.3 | 49.3 | 281.0 | 80.0 | 66.6 | 35.0 | 25.0 | 354.0 | 651.7 | — | 207.0 | 73.0 | 379,71 | 16.1 | 281.5 |
| | | Mean | 68.8 | 30.5 | 37.4 | 49.6 | 13.6 | 9.2 | 15.9 | 77.3 | 31.9 | — | 68.0 | 21.6 | 35,378 | 10.8 | 59.7 |
| | | SD | 64.8 | 14.8 | 52.3 | 12.1 | 19.2 | 8.7 | 7.3 | 83.9 | 119.4 | — | 35.7 | 22.6 | 70,515.9 | 3.7 | 87.1 |

Sc and V show the highest mean concentrations in the Maule Region (average: 33 and 225 ppm; maximum: 44 and 286 ppm, respectively). The maximum mean contents for Cr and Y are present in the Atacama region (average: 418 ppm and 73 ppm; maximum: 128,957 ppm and 4245 ppm, respectively). According to the location of ore deposits across north and central Chile, the tailings may come from the following primary sources: porphyry Cu-Mo-(Au), porphyry Au, epithermal Au, Iron Oxide-Copper-Gold (IOCG), Iron Oxide-Apatite (IOA), stratabound and skarn [45].

Southern Chile, which includes only the Aysén Region, has tailings with the highest national concentration of Zn (average: 35,378 ppm, over 500 times higher than crustal; and a maximum of 379,713 ppm), Ta (average: 32 ppm; maximum: 652 ppm) and Hf (average: 14 ppm; maximum 67 ppm). In the case of REE (La, Ce, Pr, Nd, Tb and Dy), all of them have contents above crustal; however, the difference is not excessively high. For example, La (average: 60 ppm versus 39 ppm), Ce (average: 101 ppm versus 67 ppm), Pr (average: 13 ppm versus 9 ppm), Nd (average: 52 ppm versus 42 ppm), Tb (average: 1.4 ppm versus 1.2 ppm) and Dy (average: 7.3 ppm versus 5.2 ppm). In terms of sources, the tailings situated in the Aysén Region are probably coming from skarn and epithermal deposits mainly. In contrast, Ni, Sr and Nb show under concentration compared to crustal abundance throughout the country.

Chile is dominated by Cu production; thus, if we consider this information and evaluate the wheel of metal companionality [44] from a general perspective and not only the elements assessed in this study, the critical metals with the potential to be produced as Cu by-products, from high to low probability, should be as follows: Se, Te, Mo, Co, Re, Ag, Bi, U, Au, In, Sn and Zn. The country also produces Au and Fe as primary commodities. For Au, the byproducts from high to low probability can be as follows: Ag, Cu, Zn, Sb and U. On the other hand, for Fe, the following secondary elements may be present in the same order: Ce, Nd, Pr, La, Eu, V, Sm, Gd, Tb, Dy, Mn and Y.

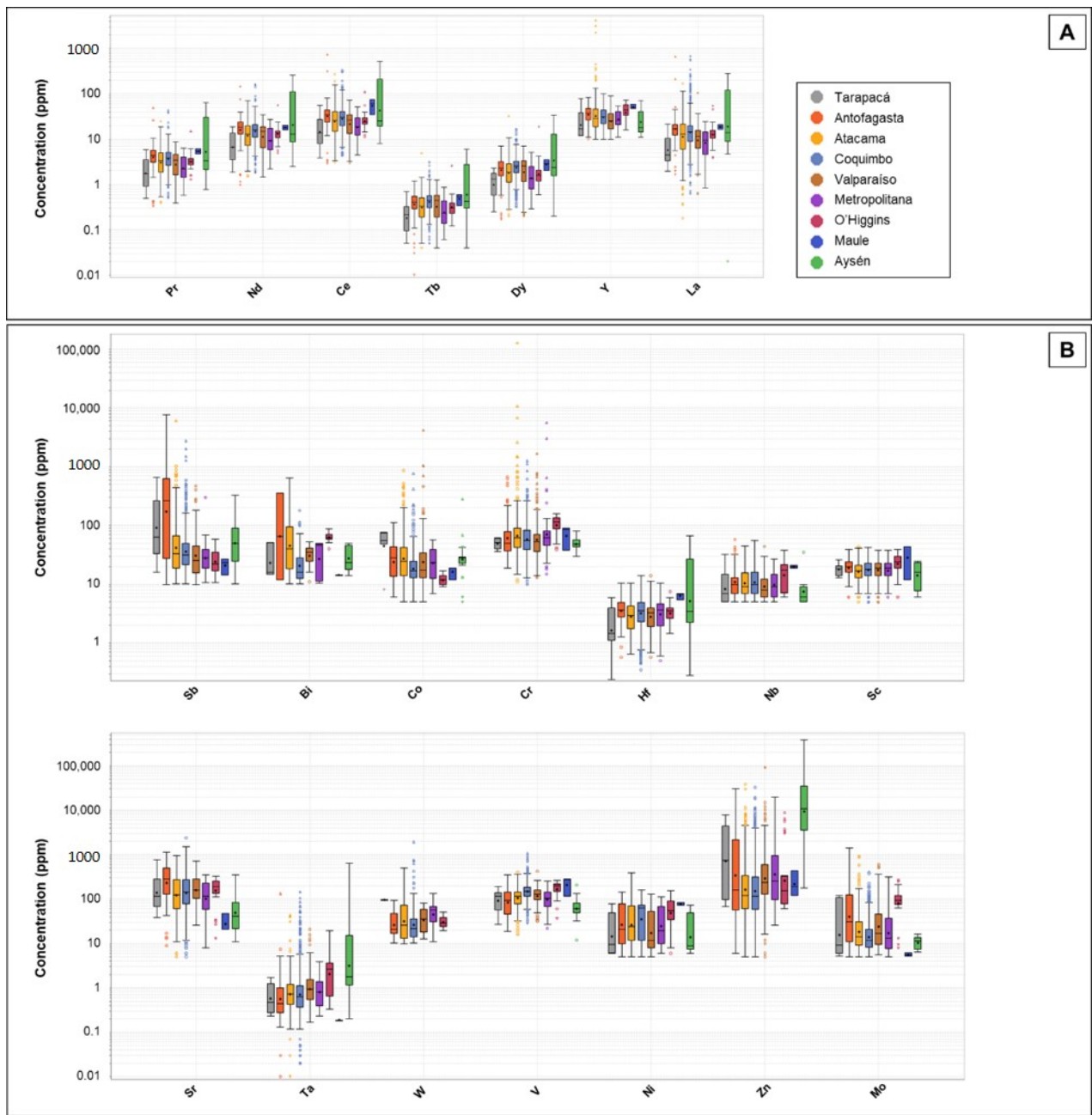

**Figure 4.** Tukey log-plot graph showing the concentration of the critical elements at each site included in Australia split in (**A**) REEs-Praseodymium, Neodymium, Cerium, Terbium, Dysprosium and Yttrium (**B**) other critical elements. The box is the middle 50% of data from Q1 to Q3. The outliers (circles) are in the top or bottom 5% of the data. The whiskers are the 5% and 95% values.

*3.7. Database Analysis for Australia*

The bulk results of select elements grouped by the site for Australia (Queensland) and their summary statistics are shown in Figure 5 and Table 5, respectively. The elements of interest from the mine waste tailings studied in this paper were Co, Cu, Ga, Ge, Hf, Li, Mo, Pb, Sb, Sc, Ta, W and Zn. The highest concentrations of Co were measured in tailings material from Rocklands (average: 561 ppm; maximum 1195 ppm) followed by Capricorn Copper (average: 77.7 ppm; maximum 1060 ppm; Figure 5). Co concentrations at all other sites are below 500 ppm. These data suggest that Co is commonly associated with IOCG and sediment hosted deposits in Queensland's Northwest Minerals Province. Cu concentrations from all sites but Horn Island are greater than 200 ppm. Rockland

tailings contain the highest copper (average: 1632 ppm; maximum 8850 ppm), followed by Capricorn Copper (average: 2032 ppm; maximum 6010 ppm). Again, this relates to the association with Cu deposits (i.e., IOCG and sediment hosted). The highest concentrations of Li were measured at Herberton (average: 60 ppm; maximum 201 ppm) whereas the highest concentrations of Sb were measured in Capricorn Copper tailings (average: 59 ppm; maximum 1220 ppm). The tailings material from Wolfram Camp contains the highest concentrations of Mo and W across the seven sites (average: 371 ppm; maximum 2429 ppm and average: 540 ppm; maximum 2728 ppm) and is likely related to the greisen style of mineralisation.

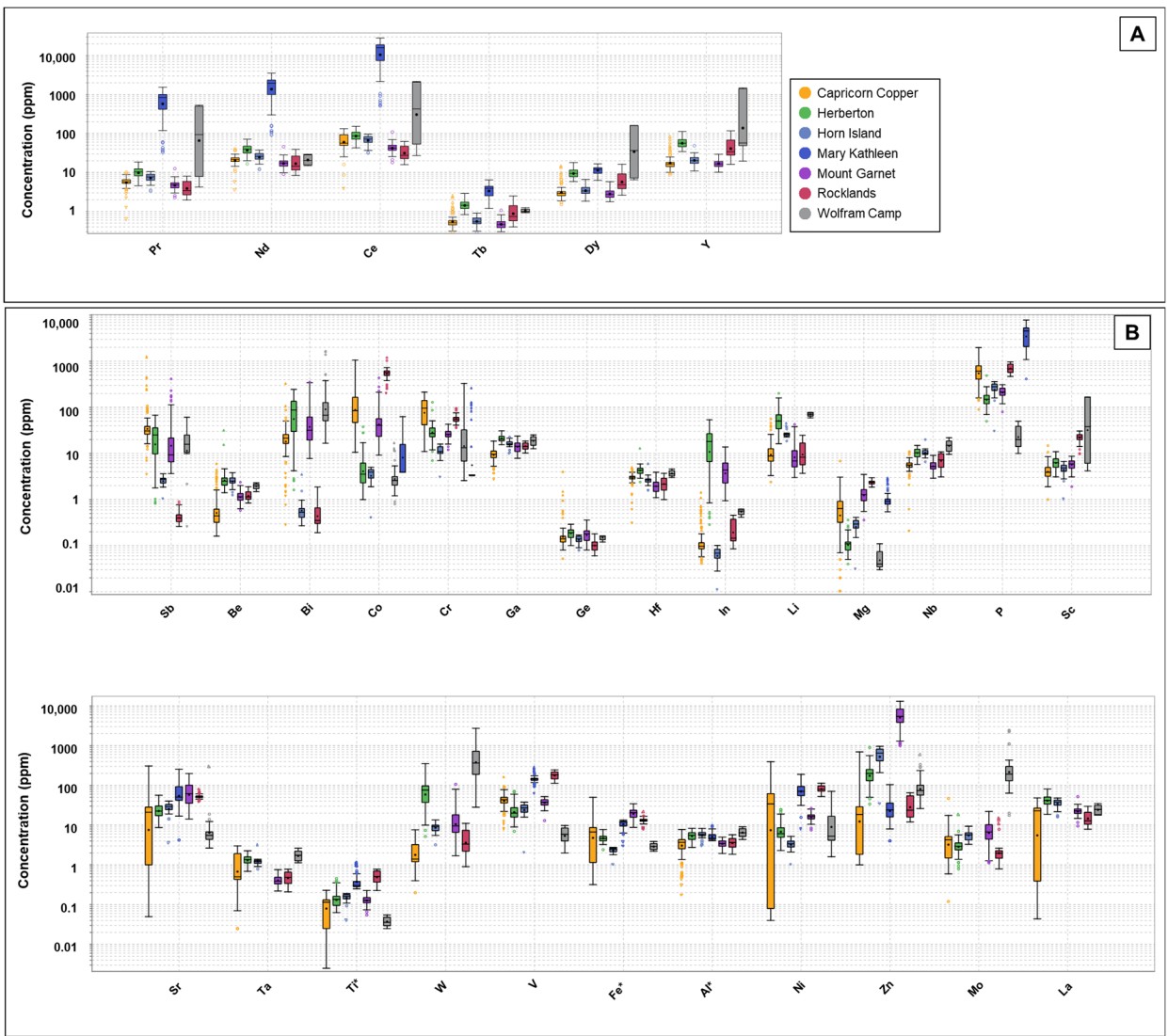

**Figure 5.** Tukey log-plot graph showing the concentration of the critical elements at each deposit included in Australia split in (**A**) REEs-Praseodymium, Neodymium, Cerium, Terbium, Dysprosium and Yttrium (**B**) other critical elements. All concentrations in ppm except Titanium, Iron and Aluminium displayed as %. The box is the middle 50% of data from Q1 to Q3. The outliers (circles) are in the top or bottom 5% of the data. The whiskers are the 5% and 95% values.

**Table 5.** Summary statistics of the elements classified as critical or with high economic relevance in each region from the Chilean data used in this study. Highlighted figures indicate a concentration above crustal abundance. SD: standard deviation; Min: minimum value; Max: maximum value.

| Element (ppm) | | Rare Earth Elements | | | | | |
| --- | --- | --- | --- | --- | --- | --- | --- |
| | | Pr | Nd | Ce | Tb | Dy | Y |
| Capricorn Copper | Min. | 0.6 | 3.5 | 3.8 | 0.3 | 1.5 | 8.8 |
| | Max. | 10.2 | 39.9 | 132.5 | 2.5 | 14.4 | 82.1 |
| | Mean | 5.7 | 21.3 | 65.8 | 0.6 | 3.4 | 19.0 |
| | SD | 1.4 | 5.2 | 26.9 | 0.3 | 2.0 | 11.2 |
| Herberton | Min. | 4.6 | 16.6 | 42.8 | 0.8 | 5.8 | 34.3 |
| | Max. | 18.6 | 71.3 | 153.0 | 2.9 | 18.0 | 112.5 |
| | Mean | 10.2 | 38.7 | 88.9 | 1.5 | 9.7 | 58.5 |
| | SD | 2.8 | 10.7 | 23.8 | 0.4 | 2.5 | 16.7 |
| Horn Island | Min. | 3.3 | 11.9 | 31.6 | 0.3 | 1.8 | 10.9 |
| | Max. | 10.5 | 37.5 | 96.1 | 0.9 | 6.6 | 48.6 |
| | Mean | 7.5 | 25.7 | 69.6 | 0.6 | 3.6 | 21.6 |
| | SD | 2.0 | 6.9 | 18.6 | 0.2 | 1.1 | 7.8 |
| Mary Kathleen | Min. | 32.9 | 91.7 | 510.7 | 1.2 | 6.3 | — |
| | Max. | 1534.8 | 3567.4 | 28,142.3 | 6.5 | 16.7 | — |
| | Mean | 774.3 | 1820.2 | 14,392.9 | 3.7 | 11.7 | — |
| | SD | 427.4 | 992.2 | 8021.2 | 1.4 | 2.9 | — |
| Mount Garnet | Min. | 2.3 | 9.0 | 18.1 | 0.3 | 1.8 | 10.2 |
| | Max. | 12.6 | 45.8 | 109.0 | 1.1 | 5.8 | 29.3 |
| | Mean | 4.9 | 17.7 | 43.6 | 0.5 | 2.9 | 17.3 |
| | SD | 1.5 | 5.3 | 13.3 | 0.1 | 0.8 | 4.2 |
| Rocklands | Min. | 2.0 | 8.4 | 15.9 | 0.4 | 2.6 | 16.1 |
| | Max. | 8.1 | 38.9 | 62.5 | 2.5 | 16.4 | 117.5 |
| | Mean | 4.2 | 18.5 | 33.9 | 1.0 | 6.4 | 46.5 |
| | SD | 1.8 | 8.3 | 13.6 | 0.5 | 3.3 | 24.7 |
| Wolfram Camp | Min. | 4.2 | 14.8 | 27.1 | 0.9 | 6.5 | 19.4 |
| | Max. | 533.4 | 29.2 | 2176.3 | 1.3 | 160.3 | 1481.7 |
| | Mean | In | 21.7 | 814.7 | 1.0 | 83.0 | 484.3 |
| | SD | 228.9 | 6.5 | 923.4 | 0.1 | 79.7 | 666.5 |

| Element (ppm) | | Others Elements with Critical and Economic Importance | | | | | | | | | |
| --- | --- | --- | --- | --- | --- | --- | --- | --- | --- | --- | --- |
| | | Sb | Be | Bi | Co | Cr | Ga | Ge | Hf | In | Li |
| Capricorn Copper | Min. | 3.4 | 0.2 | 0.3 | 10.6 | 11.0 | 2.7 | 0.1 | 0.3 | 0.0 | 2.4 |
| | Max. | 1220.0 | 5.9 | 324.0 | 1060.0 | 215.0 | 18.5 | 4.0 | 4.9 | 1.4 | 55.9 |
| | Mean | 53.2 | 0.7 | 24.4 | 142.8 | 93.3 | 9.6 | 0.2 | 3.0 | 0.2 | 11.1 |
| | SD | 30.8 | 0.4 | 21.4 | 85.1 | 96.0 | 9.7 | 0.1 | 3.1 | 0.2 | 8.9 |
| Herberton | Min. | 117.4 | 0.9 | 29.2 | 182.2 | 53.2 | 2.5 | 0.4 | 0.6 | 0.3 | 7.9 |
| | Max. | 67.7 | 31.9 | 245.0 | 37.8 | 129.0 | 31.0 | 0.3 | 12.9 | 53.7 | 201.0 |
| | Mean | 25.0 | 3.2 | 91.2 | 6.0 | 31.6 | 21.5 | 0.2 | 4.4 | 18.3 | 60.8 |
| | SD | 17.0 | 4.1 | 62.5 | 6.9 | 17.3 | 3.6 | 0.0 | 1.3 | 12.7 | 40.0 |
| Horn Island | Min. | 1.0 | 1.3 | 0.3 | 0.4 | 3.0 | 9.5 | 0.1 | 1.5 | 0.0 | 18.6 |
| | Max. | 3.6 | 3.8 | 3.5 | 5.0 | 16.0 | 22.8 | 0.2 | 5.9 | 0.1 | 44.7 |
| | Mean | 2.6 | 2.6 | 0.7 | 3.6 | 11.3 | 16.3 | 0.1 | 2.7 | 0.1 | 27.1 |
| | SD | 0.5 | 0.7 | 0.6 | 1.1 | 3.2 | 3.3 | 0.0 | 0.8 | 0.0 | 5.8 |
| Mary Kathleen | Min. | — | — | — | 3.9 | 3.4 | — | — | — | — | — |
| | Max. | — | — | — | 62.9 | 260.0 | — | — | — | — | — |
| | Mean | — | — | — | 11.6 | 18.5 | — | — | — | — | — |
| | SD | — | — | — | 10.9 | 42.9 | — | — | — | — | — |

Table 5. *Cont.*

| Element (ppm) | | Others Elements with Critical and Economic Importance | | | | | | | | | |
|---|---|---|---|---|---|---|---|---|---|---|---|
| | | Sb | Be | Bi | Co | Cr | Ga | Ge | Hf | In | Li |
| Mount Garnet | Min. | 3.7 | 0.6 | 7.8 | 9.2 | 12.0 | 7.8 | 0.1 | 1.1 | 0.9 | 3.0 |
| | Max. | 414.0 | 2.4 | 355.0 | 436.0 | 43.0 | 23.8 | 0.4 | 3.9 | 13.8 | 42.7 |
| | Mean | 34.7 | 1.2 | 59.3 | 61.0 | 26.4 | 14.4 | 0.2 | 2.0 | 4.4 | 10.4 |
| | SD | 66.1 | 0.3 | 70.1 | 69.9 | 6.3 | 3.6 | 0.1 | 0.7 | 2.5 | 8.4 |
| Rocklands | Min. | 0.3 | 0.8 | 0.2 | 199.5 | 36.0 | 10.3 | 0.1 | 1.0 | 0.1 | 3.7 |
| | Max. | 0.9 | 1.9 | 1.9 | 1195.0 | 95.0 | 18.5 | 0.2 | 3.7 | 0.5 | 24.8 |
| | Mean | 0.4 | 1.2 | 0.5 | 561.1 | 57.3 | 14.6 | 0.1 | 2.3 | 0.2 | 11.0 |
| | SD | 0.1 | 0.3 | 0.3 | 156.9 | 12.7 | 2.3 | 0.0 | 0.7 | 0.1 | 5.8 |
| Wolfram Camp | Min. | 0.3 | 1.5 | 16.7 | 0.8 | 2.6 | 12.7 | 0.1 | 3.0 | 0.4 | 59.3 |
| | Max. | 61.0 | 2.3 | 1640.5 | 16.7 | 329.8 | 25.2 | 0.2 | 4.6 | 0.6 | 77.2 |
| | Mean | 19.2 | 2.0 | 187.7 | 3.5 | 31.5 | 19.2 | 0.2 | 3.7 | 0.5 | 71.0 |
| | SD | 15.8 | 0.3 | 360.1 | 3.1 | 59.3 | 4.6 | 0.0 | 0.6 | 0.1 | 7.2 |

| Element (ppm) | | Others Elements with Critical and Economic Importance | | | | | | | | |
|---|---|---|---|---|---|---|---|---|---|---|
| | | Sr | Ta | Ti | W | V | Ni | Zn | Mo | La |
| Capricorn Copper | Min. | 0.1 | 0.0 | 0.0 | 0.2 | 7.8 | 0.0 | 1.0 | 0.1 | 0.0 |
| | Max. | 307.0 | 3.0 | 0.2 | 7.6 | 160.0 | 394.0 | 697.0 | 47.0 | 48.2 |
| | Mean | 30.9 | 1.0 | 0.1 | 2.2 | 43.8 | 39.8 | 36.7 | 4.4 | 17.5 |
| | SD | 48.7 | 0.9 | 0.1 | 1.6 | 17.2 | 44.6 | 89.5 | 4.6 | 11.9 |
| Herberton | Min. | 8.7 | 0.7 | 0.1 | 5.2 | 7.0 | 2.3 | 34.0 | 0.8 | 18.7 |
| | Max. | 56.4 | 2.3 | 0.5 | 351.0 | 71.0 | 24.8 | 901.0 | 18.2 | 81.0 |
| | Mean | 24.7 | 1.4 | 0.1 | 82.1 | 24.1 | 8.1 | 210.5 | 3.2 | 43.1 |
| | SD | 10.1 | 0.4 | 0.1 | 64.6 | 13.9 | 5.7 | 138.0 | 2.2 | 12.6 |
| Horn Island | Min. | 3.5 | 0.8 | 0.0 | 3.2 | 2.0 | 1.0 | 34.0 | 3.3 | 16.2 |
| | Max. | 40.3 | 3.1 | 0.2 | 13.5 | 38.0 | 5.2 | 959.0 | 9.4 | 48.0 |
| | Mean | 27.5 | 1.3 | 0.2 | 9.0 | 26.4 | 3.4 | 607.8 | 5.7 | 36.1 |
| | SD | 8.5 | 0.4 | 0.0 | 2.3 | 8.5 | 0.9 | 248.2 | 1.5 | 9.0 |
| Mary Kathleen | Min. | 4.2 | — | 0.2 | — | 61.6 | 7.9 | 4.0 | — | — |
| | Max. | 253.7 | — | 1.2 | — | 280.1 | 188.6 | 104.5 | — | — |
| | Mean | 74.5 | — | 0.4 | — | 151.8 | 78.4 | 30.8 | — | — |
| | SD | 61.8 | — | 0.3 | — | 41.6 | 33.7 | 25.8 | — | — |
| Mount Garnet | Min. | 14.2 | 0.2 | 0.1 | 1.7 | 13.0 | 7.6 | 1010.0 | 1.1 | 9.5 |
| | Max. | 199.0 | 0.8 | 0.2 | 106.5 | 52.0 | 26.3 | 13,050.0 | 22.0 | 52.0 |
| | Mean | 70.3 | 0.4 | 0.1 | 15.3 | 37.9 | 16.4 | 6077.2 | 7.9 | 22.9 |
| | SD | 41.0 | 0.1 | 0.0 | 17.0 | 7.8 | 3.4 | 3143.2 | 4.9 | 6.1 |
| Rocklands | Min. | 37.5 | 0.2 | 0.2 | 0.9 | 113.0 | 52.5 | 12.0 | 0.8 | 7.9 |
| | Max. | 78.6 | 0.8 | 0.8 | 11.1 | 246.0 | 113.5 | 65.0 | 14.6 | 29.6 |
| | Mean | 52.4 | 0.5 | 0.5 | 4.5 | 181.6 | 82.2 | 34.0 | 2.5 | 15.7 |
| | SD | 6.3 | 0.2 | 0.2 | 2.8 | 39.1 | 16.5 | 19.5 | 2.6 | 6.1 |
| Wolfram Camp | Min. | 2.6 | 1.1 | 0.0 | 28.1 | 2.0 | 1.6 | 26.2 | 17.6 | 17.9 |
| | Max. | 296.8 | 2.6 | 0.1 | 2728.5 | 9.8 | 70.9 | 597.3 | 2429.3 | 34.9 |
| | Mean | 17.4 | 1.8 | 0.0 | 540.8 | 6.0 | 15.7 | 106.3 | 371.3 | 25.0 |
| | SD | 57.0 | 0.6 | 0.0 | 505.7 | 2.6 | 19.6 | 102.2 | 551.2 | 7.1 |

## 4. Conclusions

This study enabled a review of state-of-the-art tailings information, characterisation and metal recovery activities currently being undertaken to Mexico, Chile and Australia and facilitated a critical review to be undertaken in order to help guide transformative potential for the three countries. The compiled database intended to assess the potential for tailings as a secondary resource of elements classified as critical or with high economic relevance in these countries. It is noteworthy that the studied countries each have different quantities

of publicly available tailings data, characterisation and recovery of critical elements from tailings (i.e., Chile n= 642; Mexico 159; Australia >6). Nevertheless, these preliminary results suggest abandoned tailings as a promising secondary source of these elements. Data analysis shows that Mexico has significant potentiality for Bi, Sb, W, In, Zn and Mo in Sonora State, while Chile has significant potential for Bi, Sb, W and Mo, mostly from northern to central regions and Zn to the south. Whilst data from Australia are still being compiled nationally, the potential for Co in Northwest Queensland, where data were sourced, was noticed. The different distribution of metals in tailings relates primarily to ore deposit types from which primary mining had targeted; thus, the differences in metal contents reported from these three countries reflect this: the difference in economic cut-off values used at each mine site and, indeed, the difference in mineralogical processing technologies used. Furthermore, these countries all experience different climates; thus, secondary cycling of metals in the tailing's storage facilities is likely and may impact the concentrations of metals measured and analysed in this study.

Research highlights that available information is insufficient and indicates that there is a need for the development of international reports or assessment code for mine waste so that these data can be systematically quantified and reported, thus enabling future mineral exploration in these new terrains. Additional valuable data that should be captured in a future iteration of this type of database could include information on physical characteristics of tailings (e.g., particle size distribution), tailings mineralogy, distribution of elements amongst different size fractions and include any information relating to potential metallurgical responses. Collectively, this information will allow developing a techno-economic feasibility analysis of extracting critical elements from mine tailings, thus providing both the socio-economic impetus and the pre-cursor to developing a business case for reprocessing to help the mining sector to join the societal transition from a linear to a circular economy that promises to achieve multiple Sustainable Development Goals.

**Supplementary Materials:** The following are available online at https://www.mdpi.com/article/10.3390/min12020122/s1, Table S1: Collected data from Mexico and Table S2: Collected data from Chile, Table S3: Characteristics of Australian tailings sites used in this study, Table S4: Detection Limits for the Australian Tailings Samples used in this study.

**Author Contributions:** Conceptualization, D.V.G. methodology, D.V.G., E.S.S., O.M., A.M.P.-E., L.A.P.T., L.J. and A.P.-F.; data analysis, E.S.S., O.M., A.M.P.-E., L.A.P.T. and L.J.; investigation, D.V.G., E.S.S., O.M., A.M.P.-E., L.A.P.T., L.J. and A.P.-F.; resources, D.V.G. and A.P.-F.; writing—original draft preparation, D.V.G., E.S.S., O.M., A.M.P.-E., L.A.P.T., L.J. and A.P.-F.; writing—review and editing, D.V.G. and A.P.-F.; supervision, D.V.G. and A.P.-F.; project administration, D.V.G.; funding acquisition, D.V.G. All authors have read and agreed to the published version of the manuscript.

**Funding:** This research was funded by the Council on Australia Latin America Relations (COALAR) COALAR Special Grant Round SGR2020-040. The Queensland Government is also acknowledged for the provision of funding through the Geological Survey of Queensland under New Economy Minerals Initiative.

**Data Availability Statement:** Not applicable.

**Acknowledgments:** The Geological Survey of Queensland Queensland Government is acknowledged for the provision of funding under the New Economy Minerals Initiative, assistance with sample collection, organising geochemical analyses and data processing. We also thank students Priyanka Mirchandani and Aranza Luna Sagredo for their assistance during data collection.

**Conflicts of Interest:** The authors declare no conflict of interest. The funders had no role in the design of the study; in the collection, analyses or interpretation of data; in the writing of the manuscript; or in the decision to publish the results.

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
