# Peer review of "Data Integration of Critical Elements from Mine Waste in Mexico, Chile and Australia"

_minerals, doi:10.3390/min12020122_

Round 1
Reviewer 1 Report
Please find comments and suggestions for authors attached in the file.

Author Response
Comments in document.

Reviewer 2 Report
The research created a database using existing tailings data from three countries, namely Chile, Mexico, and Australia. It investigated whether tailings have the potential to be used as a secondary resource of critical or high economic important elements. This is an essential and informative piece of work. I've included my comments below:
- All numbers should be rechecked and corrected through the article. For example, section 3.4. General Analysis of the Collected Data, lines 324 and 325, average concentration for Sc (i.e., 18.67 for Chile, Table 2) is not above crustal abundance (i.e., 22, Table 1). Lines 325 and 326, Y, Co, Cr, and Hf averages (for Chile) are not two to five times higher than crustal. Ta is not between 15-40 times, and so on.
- Adding a few recommendations for further studies could be beneficial. For the elements with high “potential ratio”, other information such as mineralogical composition (although there is already some information in the database for Mexico), particle size distribution, assay and distribution of different elements in different size fractions of tailings, metallurgical response (i.e., metal recovery activities),… could be recommended to be added to the database in the future.
Reviewer 3 Report
In this manuscript (Minerals-1533494), entitled “Data integration of critical elements from mine waste in Australia, Mexico and Chile”, the authors confirmed the value of establishing a chemical database from publically available tailings data collated from the three countries to assess their potential as a secondary resource of elements classified as critical or with high economic relevance, which was important to formulate international reporting and encourage the recycling of waste resources. Therefore, I suggested that this manuscript can be accepted after minor revision. The following issues should be addressed.
- The authors only summarized the tailings information and metal recovery in the three countries of Australia, Mexico and Chile. In the manuscript, the authors should describe the basis for choosing these three countries.
- The authors should establish corresponding links between the tailings resource information and mining in the area, so that it is easy to understand the sources of different types of tailings.
- The authors should provide the recyclable standards of different metal elements in the article and evaluate the market value of recycled tailings resources in order to facilitate the formulation of government policies.
- There are obvious differences in the distribution of tailings and metal element recovery in the three countries. The author can summarize the reasons for the differences and the same law of differences in different regions.
